# Metastable brain waves

James A. Roberts [1,2], Leonardo L. Gollo [1,2], Romesh G. Abeysuriya[3], Gloria Roberts[4,5], Philip B. Mitchell[4,5], Mark W. Woolrich[3] & Michael Breakspear [1,2,6,7]

Traveling patterns of neuronal activity—brain waves—have been observed across a breadth of neuronal recordings, states of awareness, and species, but their emergence in the human brain lacks a firm understanding. Here we analyze the complex nonlinear dynamics that emerge from modeling large-scale spontaneous neural activity on a whole-brain network derived from human tractography. We find a rich array of three-dimensional wave patterns, including traveling waves, spiral waves, sources, and sinks. These patterns are metastable, such that multiple spatiotemporal wave patterns are visited in sequence. Transitions between states correspond to reconfigurations of underlying phase flows, characterized by nonlinear instabilities. These metastable dynamics accord with empirical data from multiple imaging modalities, including electrical waves in cortical tissue, sequential spatiotemporal patterns in resting-state MEG data, and large-scale waves in human electrocorticography. By moving the study of functional networks from a spatially static to an inherently dynamic (wave-like) frame, our work unifies apparently diverse phenomena across functional neuroimaging modalities and makes specific predictions for further experimentation.

[1] QIMR Berghofer Medical Research Institute, Brisbane, QLD 4006, Australia. [2] Centre for Integrative Brain Function, QIMR Berghofer Medical Research Institute, Brisbane, QLD 4006, Australia. [3] Oxford Centre for Human Brain Activity (OHBA), Wellcome Centre for Integrative Neuroimaging, Department of Psychiatry, University of Oxford, Oxford OX3 7JX, UK. [4] School of Psychiatry, University of New South Wales, Sydney, NSW 2052, Australia. [5] Black Dog Institute, Prince of Wales Hospital, Hospital Road, Randwick, NSW 2031, Australia. [6] Metro North Mental Health Service, Royal Brisbane and Women's Hospital, Brisbane, QLD 4029, Australia. [7] Present address: Hunter Medical Research Institute, University of Newcastle, Newcastle, NSW 2305, Australia. Correspondence and requests for materials should be addressed to J.A.R. (email: james.roberts@qimrberghofer.edu.au)

A central aim in neuroscience is to understand how complex brain dynamics emerge from brain structure. Thus far, attention has been largely directed toward understanding long-time-averaged measures of brain activity such as correlations and power spectra—i.e., static summaries of the underlying dynamics. Although the importance of dynamics has long been known in electrophysiology experiments, human neuroimaging has been slow to embrace this additional information. However, this is rapidly changing due to advances in imaging technology, such as fast functional magnetic resonance imaging (fMRI), and accompanying analysis methods. Time-varying analyses have recently revealed richer dynamics than previously appreciated: brain activity exhibits switching between metastable states[1], stochastic jumps between multistable states[2], transiently expressed functional networks[3], and large-scale waves[4]. Yet, with the exception of the pathological strongly nonlinear dynamics in epileptic seizures[5], theory and modeling have fallen behind the body of empirical results. Various empirical debates have appeared in this vacuum, such as the existence and nature of non-stationary (dynamic) functional connectivity (FC)[3,6]. Models with physiologically meaningful parameters have much to contribute here, through their ability to specify candidate causes of complex patterns in empirical data[7].

Waves are canonical examples of dynamical phenomena in biological systems. A diversity of neuronal wave patterns have been observed on mesoscopic[8–12] and whole-brain scales[4,13–15]. These waves are not merely epiphenomena: e.g., they have been reproducibly observed in visual processing[16]—carrying the primary stimulus-evoked response in visual cortex[8,17]; reflecting information flow in response to dynamic natural scenes[18]; encoding directions of moving stimuli[19]; encoding stimulus positions and orientations[20]; underlying bistable perceptual rivalry[21]; reinforcing recent visual experience[22]; and also occur pathologically during visual hallucinations[23]. Waves have been observed in primary motor cortex, where they mediate information transfer during movement preparation[9], can be induced by optogenetic stimulation[24], and reveal the nature of the excitability of neural tissue[25]. They have also been implicated in sensorimotor processing of saccades[26], propagating seizure fronts[27,28], and observed during sleep[29] with a possible role in memory consolidation[4]. Waves have been reported in diverse neuroimaging modalities including voltage-sensitive dyes (VSDs)[8,17,22,30–32], local field potentials[9,12,18,19,26], electro-corticography[4,29], electroencephalography (EEG)[13,14], magnetoencephalography (MEG)[33], and fMRI[21], and inferred from close analysis of psychophysical phenomena[34]. The widespread occurrence of cortical waves opens many questions:[15] What is their basis? Is each instance a uniquely determined phenomenon or do there exist deeper unifying principles? How do particular waves appear and disperse, and how do they relate to stationary patterns of activity? Computational models are required to tackle these questions.

The linear treatment of cortical waves under idealized assumptions regarding cortico-cortical connectivity has been well-studied[35], particularly for standing waves[36,37], as has neuronal wave pattern formation in abstract mathematical settings[23,38–40]. However, large-scale waves and spontaneous transitions between different emergent patterns in models of brain dynamics constrained by empirical connectivity data have not been explored. Here we show that large-scale metastable waves emerge and dynamically evolve on the human connectome, whereby the system visits multiple patterns in sequence and no single wave pattern endures. Importantly, the presence of these waves does not depend upon the choice of neural model and is replicated on two independent whole-brain connectomes.

## Results

**Network model.** We modeled large-scale brain dynamics using a network of coupled neural masses[41–45]. This approach has two main components: a local mean-field model that describes neuronal dynamics in each region and a structural connectome that introduces connectivity between regions[7]. We describe the local dynamics of each brain region with a conductance-based neural mass[46]. We first concentrate on dynamics in the absence of noise.

Regions are coupled by connections between the excitatory populations. We use connectomic data derived from healthy subjects using probabilistic tractography[47]. This yields the connectivity matrix, which describes direct connections between regions and the strengths of these connections. A global coupling constant $c$ scales all the connection weights and operates as a tuning parameter that sets the overall excitability of the brain[48]. We also include delays between regions, which are important[41], particularly for inter-areal synchronization properties[43,49]. As a first approximation, we use a constant delay $\tau$ for all connections.

The key novel model ingredients here are that we (i) explore a range of coupling strengths and delays beyond the narrow area of parameter space previously studied; (ii) use higher-quality and denser connectivity data than previous modeling studies[44]; and (iii) focus on spatiotemporal dynamics unfolding on a wide range of time scales including the very short, not just long-time averages.

**Emergent wave dynamics.** We choose a coupling strength higher than has typically been explored previously[43,44,49]. Along with short delays, this strong coupling means that each region exerts a strong influence on its neighbors, which tends to favor local synchronization[46]. Starting from a broad diversity of random initial conditions, we find that cortical activity reliably and rapidly self-organizes into spatiotemporal patterns (Fig. 1). There appears a diversity of patterns of wave dynamics: traveling waves (Fig. 1a, Supplementary Movie 1), rotating (or spiral) waves (Fig. 1b, Supplementary Movie 2), and sources and sinks (or breathers) in which activity either emanates from or converges toward a localized point (Fig. 1c, Supplementary Movie 3). The wave patterns are highly coherent across the cortex, to the extent that most regions contribute to any given pattern. This large spatial scale is consistent with waves observed in human sleep spindles[4] and slow-wave sleep[29].

Propagation speeds represent a succinct and testable attribute of such predicted waves. Wave propagation speeds (see Methods) collated over all wave patterns and brain regions show an approximately lognormal distribution (Fig. 2a). Speeds across all regions and time points in our model have a median of 26 m s$^{-1}$, although they vary widely with 10th percentile at 14 m s$^{-1}$ and 90th percentile at 64 m s$^{-1}$. We next asked whether the speeds are homogenous across the brain. To test this, we calculated the mean speed in each brain region across all wave patterns. Nodal mean speeds range from 23 to 52 m s$^{-1}$ (Fig. 2b), slower than the nodal maximum speeds and considerably narrower than the full set of instantaneous speeds. Nodal speed is associated with node degree such that hubs tend to support lower speeds than peripheral nodes ($r = -0.17$, $p = 8.1 \times 10^{-5}$; Supplementary Fig. 1), consistent with heterogeneity of time scales found in prior work[43]. Spatially, the main trend is that nodal speeds are roughly bilaterally symmetric and increase with distance from midline (Fig. 2c). However, there is additional spatial structure (Fig. 2d). The regions supporting faster wave fronts form a roughly contiguous zone in each hemisphere, spanning from the frontal lobe to the occipital lobe, via the parietal lobe and posterior aspects of the temporal lobe. Slower regions primarily occur on the midline and toward the anterior pole of the temporal lobe.

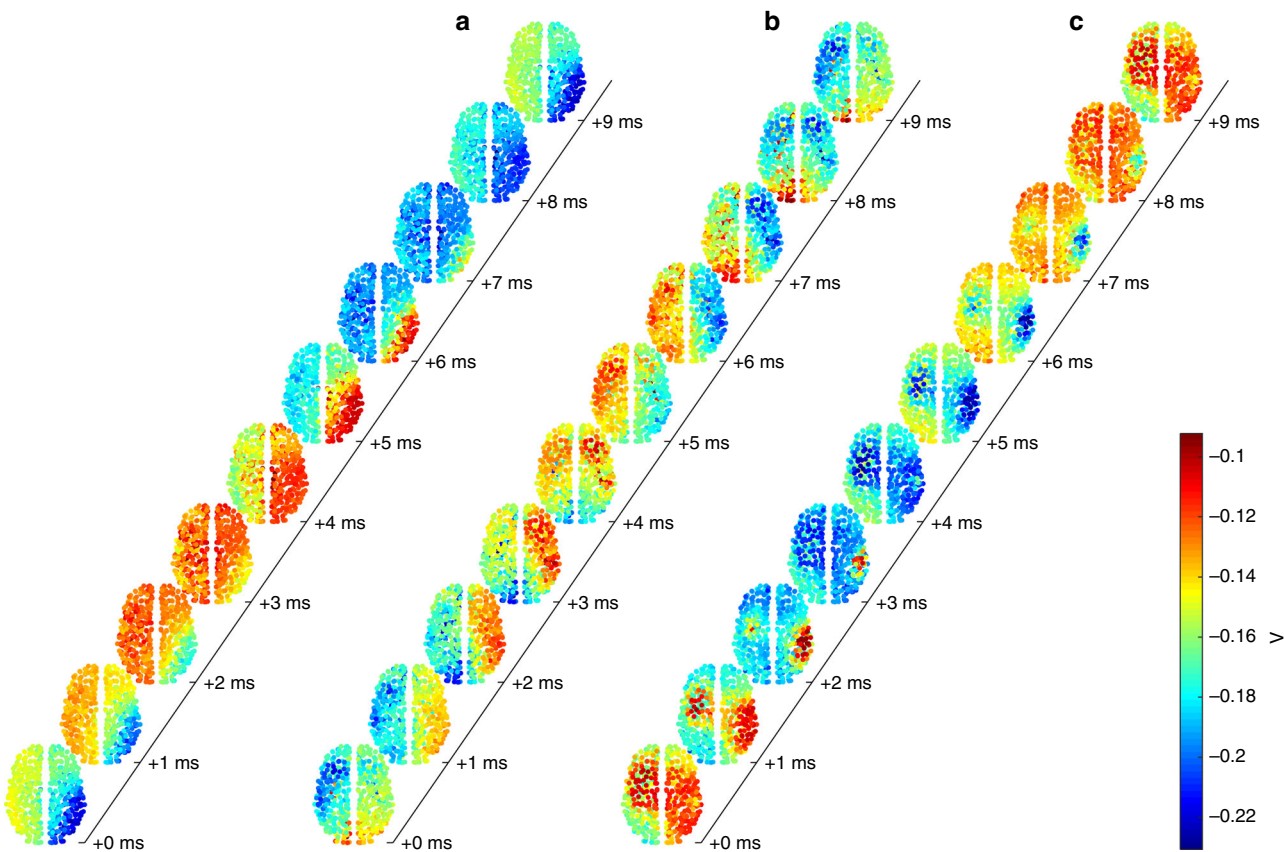

**Fig. 1** Large-scale wave patterns in the model. Ten snapshots of the dynamics of the pyramidal mean membrane potential $V$ at latencies indicated on the time axes, for **a** a traveling wave, **b** a rotating wave, and **c** a pattern with sinks (red areas shrinking for latencies 0–3 ms) and diffuse sources (broad red areas emerging for latencies 6–9 ms). These results are for strong coupling $c = 0.6$ and short delay $\tau = 1$ ms

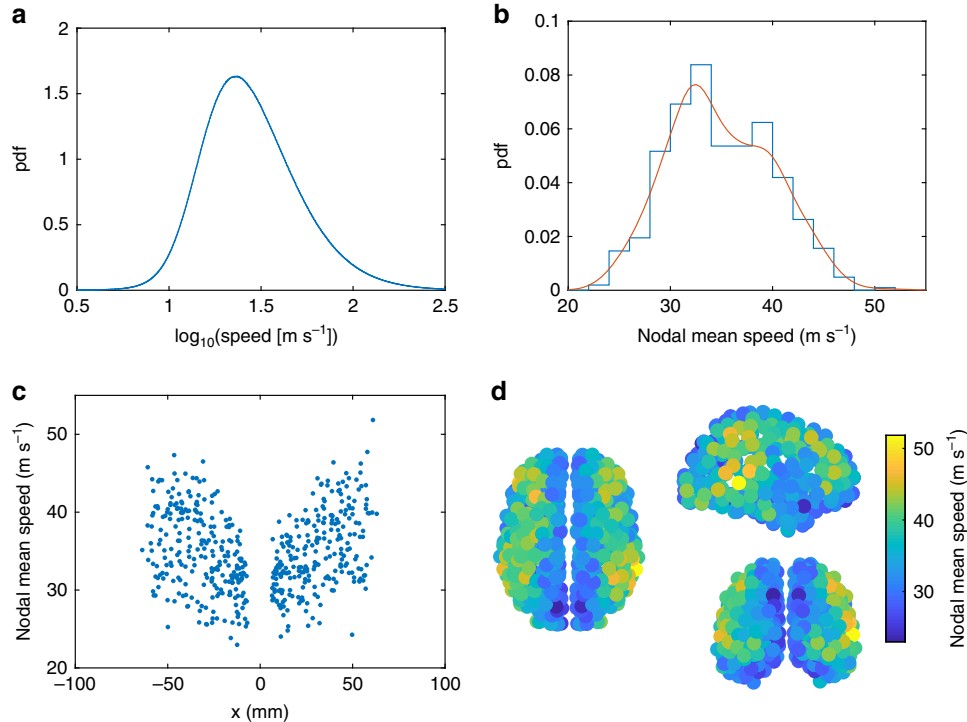

**Fig. 2** Wave propagation speeds. **a** Histogram of $\log_{10}$(speed) across all nodes and times. **b** Histogram of average speed (in m s$^{-1}$) at each node. Red line shows a kernel density estimate. **c** Mean speed in each region as a function of the lateral distance from the midline ($x = 0$). **d** Spatial distribution of nodal mean speeds as viewed (clockwise from left) from the top, right, and back. These results are for strong coupling $c = 0.6$ and short delay $\tau = 1$ ms, as in Fig. 1. Source data are provided as a Source Data file

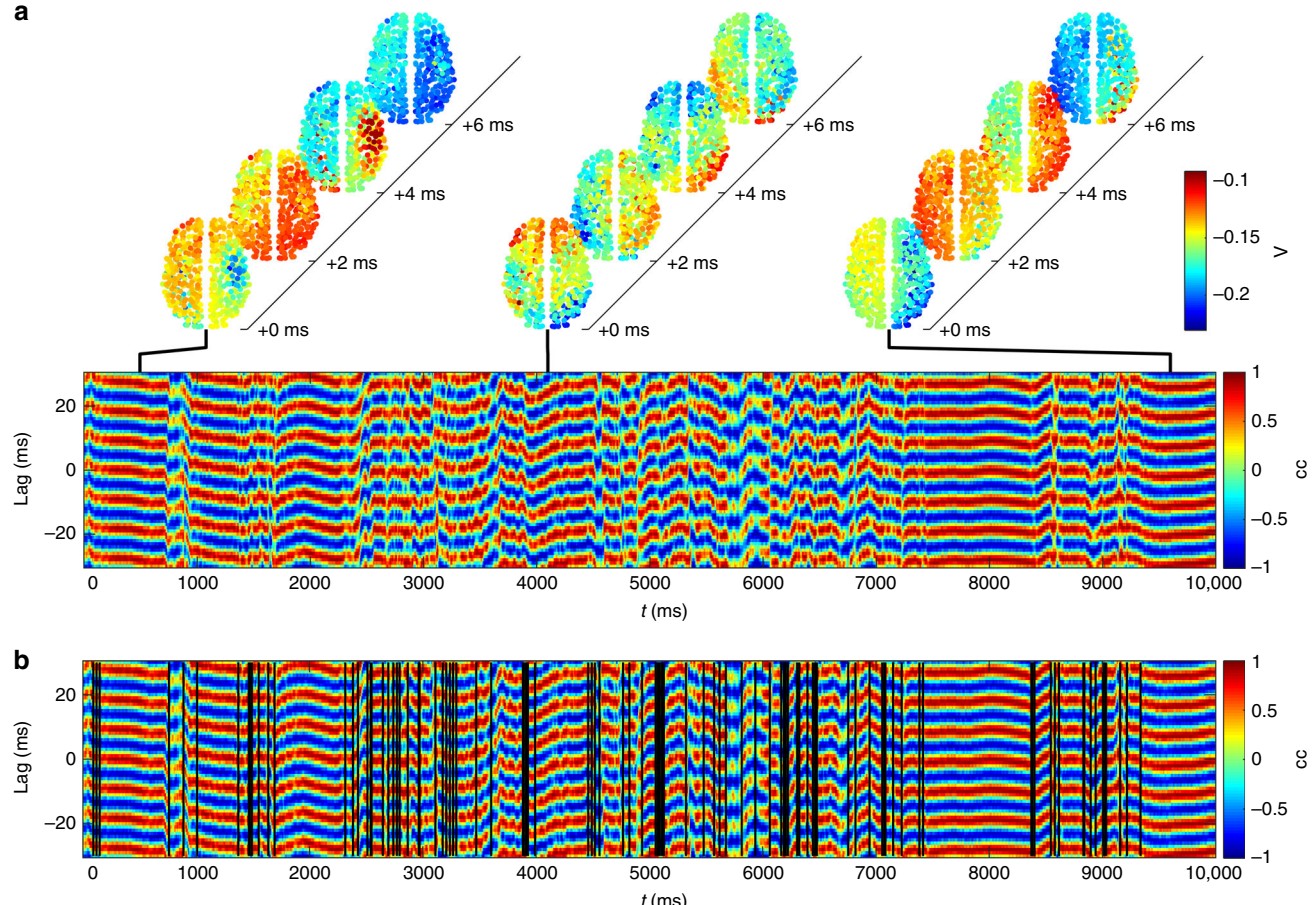

**Fig. 3** Metastable transitions. **a** Each pattern has a signature in the interhemispheric cross-correlation, with transitions between different patterns revealing a brief period of desynchronization. **b** Vertical lines depict instances of low values of the interhemispheric cross-correlation function, corresponding to wave transitions. These results are for strong coupling $c = 0.6$ and short delay $\tau = 1$ ms, as in Fig. 1

The bimodal nature of the speed distribution (Fig. 2b) suggests that slow and fast regions partition into two somewhat distinct clusters.

To place this in terms of the classic functional networks, we assigned brain regions to 12 subnetworks according to a broadly used functional subdivision of the brain[50] (Supplementary Fig. 2). The top ten fastest nodes lie in the somatomotor hand, auditory, default mode, fronto-parietal, and ventral attention networks. The top ten slowest nodes lie also in the default mode (thus indicating a wide diversity in its wave speeds), plus memory and visual regions.

**Metastable transitions**. The observed diversity of types of wave patterns (i.e., traveling waves, rotating waves, and sources and sinks) occurs for a fixed set of parameters—the dynamics shown in Fig. 1 transition spontaneously between different patterns (Supplementary Movie 4). The system dwells in a single wave pattern for many repeats of a particular wave oscillation, then exhibits a relatively rapid reconfiguration into the next pattern. These are spontaneous transitions that occur in the absence of noise or other external inputs. This rules out multistability as a mechanism for the transitions, which requires the application of a perturbation to kick the system between attractors[2,51]. Instead, what we observe is metastability, a form of winnerless competition whereby the system's orbits visit multiple patterns in sequence and no single pattern endures[51].

To quantify these metastable transitions, we use the fact that any particular wave pattern is composed of specific phase

relationships that vary relatively smoothly across space and time. The waves we observe have long wavelengths on the whole-brain scale; thus, signals averaged over a large area of cortex typically do not cancel out, as would be expected if short incoherent wavelengths dominated. To capture a metric of these patterns, we hence partition the brain into the two hemispheres and calculate the instantaneous coherence within each hemisphere (see Methods). We then calculate the sliding-window, time-lagged cross-correlation between these two intrahemispheric coherences. We term this the interhemispheric cross-correlation function.

As a particular wave pattern propagates across the brain, this pattern of correlated phase lags between the hemispheres is relatively constant. To see this, notice that during a metastable pattern (Fig. 3a), the same signature (alternating blue and red as a function of lag) persists on the time scale of hundreds of milliseconds, varying relatively slowly in time within any individual pattern (as shown by the way the blue and red stripes evolve slowly). At the time of a metastable transition, the large-scale wave pattern breaks up and disorganized short wavelengths dominate. Thus, the metastable wave signatures are separated by narrow periods of time with relatively low correlation between the hemispheres. That is, metastable transitions exhibit a brief desynchronization during which wave patterns reconfigure.

Here we used an interhemispheric partition, but any partition can be used in principle. We additionally tested partitions along the anteroposterior and dorsoventral axes, and found that they also capture the transitions (Supplementary Text, Supplementary Fig. 3).

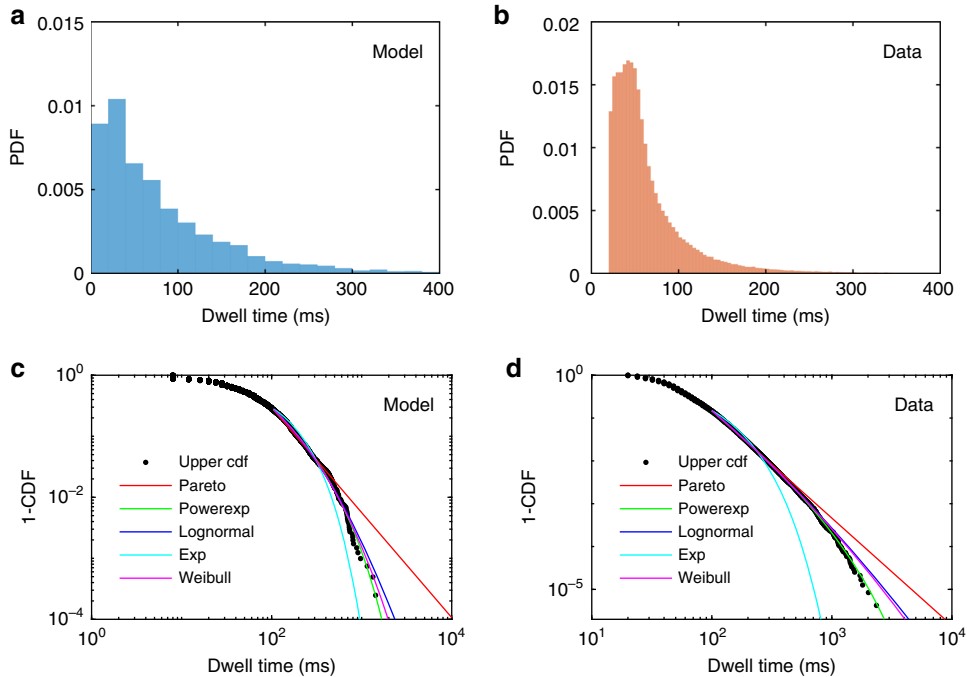

**Fig. 4** Dwell-time distributions. **a** Model for $c = 0.6$, $d = 1$ ms. **b** Resting-state MEG data from ref. [52]. **c**, **d** Upper cumulative distributions (black circles) on double logarithmic axes for dwell times in **c** the model and **d** MEG data. Lines are maximum likelihood fits to upper tails for the power law (Pareto, red), exponentially truncated power law (green), lognormal (blue), exponential (cyan), and stretched exponential (Weibull, magenta) distributions. Tail cutoffs at dwell times of 100 ms. Source data are provided as a Source Data file

We use these brief periods of phase desynchronization, yielding low values of the interhemispheric cross-correlation function, to identify transitions: This is achieved by thresholding the cross-correlation function when it is close to zero for all time lags (vertical black lines in Fig. 3b; see Methods). Extracting all times corresponding to low synchronization thus yields an automatically computed set of transition times, from which we can calculate the distribution of dwell times (Fig. 4a). We find that dwell times follow a skewed unimodal distribution.

Visual inspection of the wave dynamics shows that the same classes of waves (traveling, rotating, etc.) frequently reappear, but the precise directions of propagation and spatial configurations appear to vary. Do specific patterns recur[4,31]? To test this, we calculated the alignment of each node's velocity field with all of its past and future states. We hence find that wave patterns frequently reappear (Fig. 5a). Moreover, they appear to do so more strongly than for linear surrogate time series (Fig. 5b). To statistically test for such recurrences, we generated recurrence plots from an ensemble of linear surrogate time series (which preserve the amplitude distribution and linear spectra and cross-spectra but destroy nonlinear structure) and compared the ensuing distribution of whole-brain recurrence values with the empirical time series (Fig. 5c). Doing so confirms the substantially larger (correlated and anti-correlated) recurrences of spatiotemporal wave patterns and, through statistical thresholding, permits formal identification of when these occur (Fig. 5d).

**Comparison with resting-state data.** Transitions between dynamic spatial patterns have been recently reported in empirical resting-state MEG data[1,52]. Dwell-time distributions in these data were calculated as the times between transitions from consecutive spatial patterns in a hidden Markov model with 12 states[52]. As in our model, these dwell-time distributions exhibit skewed unimodal distributions (Fig. 4b). The individual states have median dwell time 48 ms, similar to the 56 ms median dwell time found

in the model ($p = 0.064$, Wilcoxon rank-sum test). More importantly, the distributions are of a very similar shape in their upper tails. To show this more clearly, we plot the upper cumulative distributions (Fig. 4c, d). Both distributions exhibit heavier tails than simple Gaussian or exponential distributions (cyan lines), but have thinner tails than a power law. Both exhibit good agreement with exponentially truncated power laws (green), particularly in the data where the truncated power law outperforms the alternative candidate fits (two-sided $p = 1 \times 10^{-5}$ and $p = 4 \times 10^{-6}$ for truncated power law vs. lognormal and stretched exponential, respectively, using Vuong's test). In the model, although the exponentially truncated power law again yielded the best fit, similar quality fits were found for lognormal and stretched exponential distributions (two-sided $p = 0.067$ and $p = 0.14$ for truncated power law vs. lognormal and stretched exponential, respectively, using Vuong's test). Our model thus provides a plausible mechanism for the dwell times of these metastable transitions.

Although the waves observed in our model are highly dynamic, they nonetheless bear a time-averaged signature. We thus asked what our metastable waves would look like through the standard lens of static resting-state FC as studied in fMRI. The metastable states here are relatively long-lived (mean lifetime 89 ms) but still much shorter than the typical temporal resolution of fMRI. Despite this temporal mismatch, there is empirical evidence that short-lived states visible in electrophysiology do leave an imprint on the correlations between regions at time scales visible to fMRI[1]. Calculating FC in this way on our modeled time series, after convolution with a hemodynamic response function, reveals that edgewise FC values in the model correlate with those in long-time resting-state fMRI averaged over the same group of subjects both without ($r = 0.30$, 95% confidence interval (CI) [0.293, 0.303], two-sided $p < 10^{-15}$) and with ($r = 0.41$, 95% CI [0.409, 0.418], two-sided $p < 10^{-15}$) global signal regression (GSR). This agreement compares favorably with that established in other biophysical models ($\sim r = 0.1$–$0.5$)[44,53,54], which is particularly

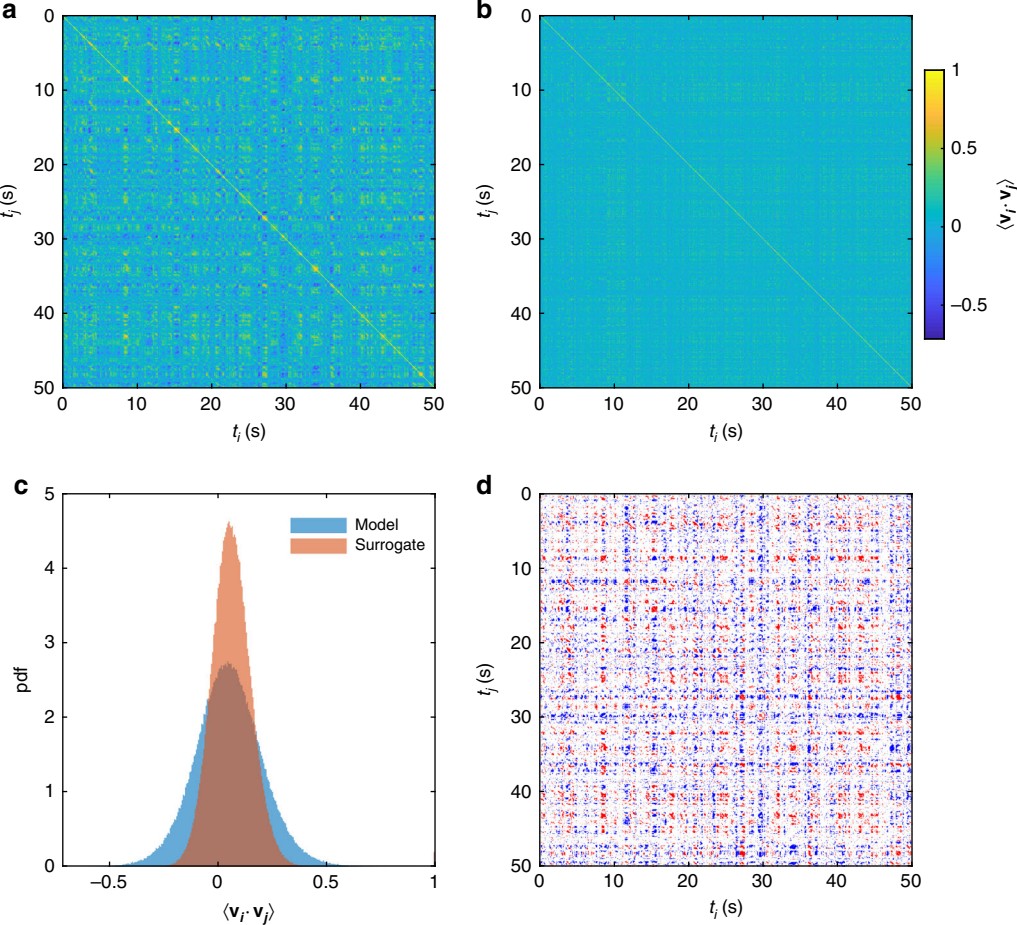

**Fig. 5** Recurring flow patterns. **a** Recurrences between the mean alignment of velocity fields $\mathbf{v}_i$ and $\mathbf{v}_j$ at times $t_i$ and $t_j$, respectively, in the model wave dynamics for $c = 0.6$, $\tau = 1$ ms. **b** Recurrences for one instance of an amplitude-adjusted Fourier surrogate time series derived from the simulation used in **a**. **c** Histograms of recurrence values for the model (blue) and one surrogate (red). **d** Recurrence points where the model recurrence alignment was greater than (red) or less than (blue) all 100 surrogates. Red points correspond to flows aligned in the same direction, whereas all blue points correspond to flows aligned in opposite directions

notable given that our finely parcellated network has an order of magnitude more regions and two orders of magnitude more connections than those typically used to study whole-brain dynamics. In addition, we compared the model FC with FC derived from MEG data. We calculated $\alpha$-band (8–13 Hz) amplitude–envelope correlations from our model time series, following a recent study[45]. Again we found that the model FC correlated with the empirical FC ($r = 0.34$, 95% CI [0.31, 0.38], two-sided $p < 10^{-15}$). Thus, through the faster lens of MEG amplitude correlations, our metastable waves again recapitulate the FC observed empirically.

It is noteworthy that here we have not tuned parameters to optimize these relationships. Instead, this shows that when viewed on the long time scale of average FC, complex wave dynamics have similar explanatory power to other model mechanisms that have been used to link brain network structure to FC. It is also worth noting that the shared structural connectivity between the model and the fMRI data would likely contribute to this correlation[44,53].

**Spatiotemporal scaffold.** We next sought to quantify the wave dynamics in a manner that would allow a better understanding of the dynamic processes underlying the wave transitions. To achieve this, we exploited the fact that the velocities form a time-varying vector field (Fig. 6a, Supplementary Movie 5), which

evolves more slowly than the waves themselves (compare Fig. 6a middle to Fig. 6a left). That is, for any specific wave pattern, the vector field is almost invariant, while the waves themselves evolve. Transitions between distinct wave patterns coincide with reconfigurations of this vector field. We hence treated the velocity vector field at each time as a snapshot of the flow of activity implied by the waves. To infer the instantaneous flow, we employed a streamline algorithm to trace the paths along the flow vectors, similar to the methods underpinning tractography using tensor-based diffusion imaging data. We calculated the streamlines in both the forward and backward directions (by inverting the flow vectors). This reveals sinks where the flow congregates and sources where the flow emerges.

Doing this reveals that streamlines typically exhibit a complex spatial arrangement, with multiple dense areas—these are the sources and sinks (Fig. 6a, right). When patterns exhibit stable traveling wave or breather solutions, the sinks and sources are typically isolated single points reflecting the dominant flow of the waves. Even though we have not imposed any temporal smoothness in the velocity or streamline estimation (e.g., using filters or regularization as done in optical flow methods[12]), the streamlines are well-behaved across time, reflecting the stability and order of the underlying activity patterns. In the example shown here, there are sources along the midline and sinks positioned laterally in both hemispheres. Occasionally, the flow

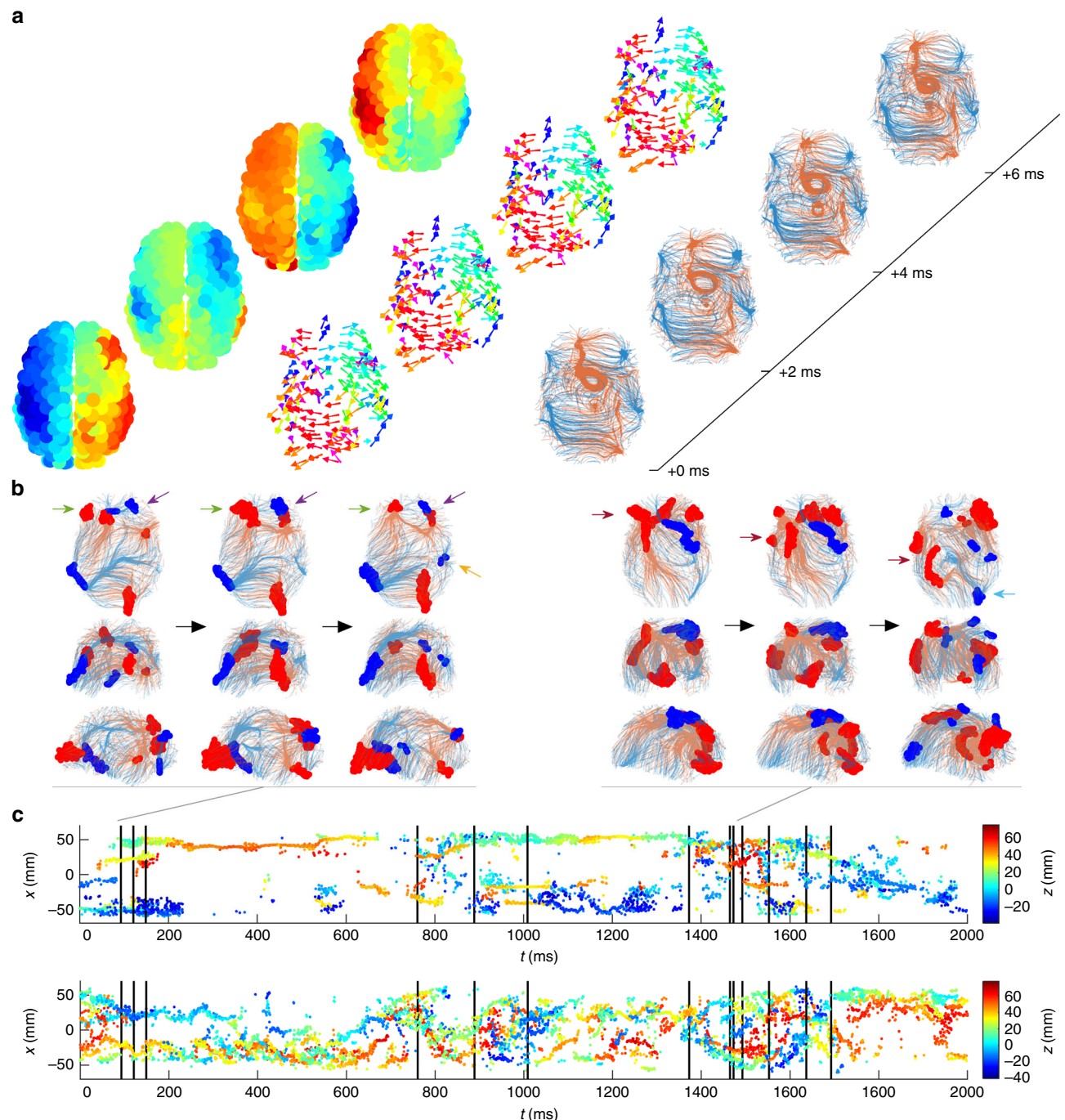

**Fig. 6** Phase flow tracked across space and time. **a** Snapshots of waves (left, colored by voltage as in Fig. 1), the corresponding phase flow vectors (middle, colored by orientation in the 2D plane shown), and phase flow streamlines (right, blue and red denote forward and backward streamlines, respectively). **b** Exemplar streamlines near metastable transitions, colored as in **a**, viewed from the top, back, and right (rows 1–3, respectively). Shown are two sets of three snapshots, each surrounding a transition as indicated (gray) in the panel below. Highlighted points (filled circles) denote clusters that form sources (red) and sinks (blue). Black arrows denote the progression of time; colored arrows denote the features referred to in the text. **c** Lateral positions (displacement from the midline, x) of sinks (top) and sources (bottom) plotted across time, colored by vertical (dorsoventral) position z. Vertical black lines denote transition times calculated using interhemispheric cross-correlation

converges onto or diverges from a closed loop (e.g., the source loop on the midline), corresponding to rotating wave patterns. The streamlines also form bundles, where nearby trajectories bunch together on approach to a sink (or from a source), revealing major pathways of activity flow.

Similar to the stable and unstable fixed points in a dynamical system, the sinks and sources succinctly summarize the dynamics

of a wave pattern. They essentially form a scaffold around which the dynamics are organized. We identify these points by finding where streamline points form dense clusters, after transients (see Methods). The sources and sinks are relatively stable on time scales of ~100 ms, with these stable periods punctuated by relatively rapid reconfigurations in space (Supplementary Movie 6). We find that metastable transitions typically coincide

with the abrupt dissolution or collision of one or more sources and/or sinks. Two exemplar metastable transitions are shown here. The first (Fig. 6b, left) shows two frontal left-hemisphere sources and a sink that merge and reconfigure to yield one left frontal source (green arrows), whereas in the right hemisphere one frontal source and one frontal sink move closer together (purple arrows), and a temporal sink emerges (yellow arrow). The second example (Fig. 6b, right) exhibits complex frontal clusters of sources and sinks that split and move posteriorly (dark red arrows), plus the emergence of an occipital sink (light blue arrow). Tracking all sources and sinks over time suggests that these reconfigurations typically coincide with desynchronizations in the interhemispheric cross-correlation function (black lines in Fig. 6c denote the same times as in Fig. 3b). To quantify this, we estimated the moment-to-moment temporal variability of sources (and sinks) by calculating the SD of the number of visits of sources (and sinks) to each node and averaged this across nodes. We used short sliding windows (non-overlapping of length 20 ms) to specifically detect rapid reconfigurations. Windows containing a cross-correlation-derived transition exhibit 14–20% higher temporal variability than those without a transition (medians for sources: 0.018 vs. 0.015, $p < 10^{-15}$; medians for sinks: 0.024 vs. 0.021, $p < 10^{-15}$; two-sided Wilcoxon rank-sum tests). We also observe that sinks tend to be more localized and less variable in time than the sources, which are often diffuse. The differences in temporal variability are highly significant (medians for sources vs. sinks, 0.022 vs. 0.016, $p < 10^{-15}$, two-sided Wilcoxon rank-sum test). The more erratic nature of the source locations is evident when comparing the upper (sinks) and lower (sources) panels of Fig. 6c. Moreover, there is a tendency for sources and sinks to remain within a single hemisphere (i.e., the trajectories traced by colored dots in Fig. 6c tend not to cross $x = 0$). This may reflect the relatively weaker interhemispheric connectivity acting as a barrier to sources and sinks traversing the hemispheres, or because the relatively small aperture causes them to collide. It could also be that slowing toward the midline means they dissolve before they get a chance to cross.

In sum, the positions of sources and sinks reflect the nature of the wave pattern at that instant in time. Metastable transitions between wave patterns correspond to their dissolution, emergence, or collisions, analogous to bifurcations in low-dimensional dynamic systems.

**Wave sources and sinks are distributed heterogeneously**. We next determined how the organizing centers of the waves are spatially distributed across the cortex. Similar to the mean speeds in Fig. 2d, the sources and sinks are heterogeneously distributed (Fig. 7a, b). Although this is to be expected to an extent, because singularities of the flow tend to occur where flow is zero (cf. the center of a spiral wave in two-dimensional (2D)), we also find regions that are highly visited by sinks but have relatively fast average flow. Sinks primarily cluster in frontal and lateral parietal areas (Fig. 7a), whereas sources are more diffuse, occurring more frequently in midline and temporal areas (Fig. 7b). This preference for midline sources may partly underlie the distribution of nodal speeds, which are typically low in the same regions (Fig. 2d), because wave fronts propagate slowly in the vicinity of a source and gather speed further away.

Do these preferential sites of sources and sinks overlap with canonical functional subnetworks (that possess strong internal FC)[50]? Both sinks (Fig. 7c) and sources (Fig. 7d) overlap non-uniformly with these functional networks (sinks: $F(12,500) = 3.5$, $p = 5.7 \times 10^{-5}$; sources: $F(12,500) = 3.7$, $p = 1.9 \times 10^{-5}$, analysis of variance; Supplementary Table 1). For example, although there is substantial overlap, the ventral attention subnetwork is visited

by sinks on average more often than each of cingulo-operculum and subcortical, whereas sources visit the somatomotor hand network more often than others including the default mode.

We also studied how these waves were shaped by the network properties of the structural connectome, finding a positive but relatively weak correlation between node strength and visits to sources ($r = 0.098$, 95% CI [0.012, 0.18], $p = 0.026$) but not sinks ($r = -0.025$, 95% CI [− 0.11, 0.061], $p = 0.57$). That is, stronger hubs are more likely to be sources than nodes that are more topologically peripheral. To further elucidate the role of hubs we compared the top 75 nodes by strength (the strongest hubs) with the bottom 75 nodes by strength (the non-hubs) and the middle feeder 363 nodes. Although there is substantial overlap, this analysis confirmed that the feeder nodes act as sinks significantly more often than hubs and non-hubs (for details, see Supplementary Table 2; Fig. 7e), whereas the hubs act as sources significantly more often than the non-hubs (Fig. 7f).

**Robustness to varying parameters, connectivity, and models**. We explored the robustness of the wave patterns to changes in the model and network details. First, we explored how the model dynamics vary over a range of coupling strengths and delays. Our aim here was to ensure that waves are not rare (confined to a unique combination of parameters) and to link the present findings to previously studied dynamical regimes. The diversity of patterns across parameter space is complex (Supplementary Movie 7). The broad differences are captured by FC matrices (Fig. 8a). We find four main classes of dynamics: waves, discrete clusters, near fully synchronized states, near fully desynchronized states, as well as hybrids of these. Broadly speaking, waves exhibit much stronger intrahemispheric FC than interhemispheric FC— or equivalently stronger short-range than long-range correlations —yielding a clear block diagonal structure (e.g., the $c = 0.6$, $\tau = 1$ ms case studied in detail above). Wave patterns (Supplementary Movie 8) can be relatively smooth (as above) or less coherent (e.g., for $c = 0.5$, $\tau = 1$ ms) such that the longest spatial scales exhibit wave patterns but within these waves, the phases are only partially synchronized between spatial neighbors. Waves are also primarily observed for stronger coupling combined with longer delays, occurring across relatively large regions of parameter space (e.g., for $c = 0.4–0.6$, $\tau = 6–10$ ms). In contrast, discrete (non-propagating) clusters are observed for weaker coupling (e.g., for $c = 0.1$, $\tau = 0, 3, 5$ ms). These discrete clusters exhibit a marked bimodal distribution of FC values, where nodes within a cluster exhibit FC $\approx 1$ (dark red), while between clusters FC$\approx 0$ (green). In the discrete cluster regime, nodes split into phase-locked clusters, with each cluster being activated sequentially (Supplementary Movie 8), as explored in detail previously[43,44,49]. A hybrid of waves and clusters, which we term lurching waves, exists for strong coupling and delays of 6–9 ms (Supplementary Movie 8). Within this regime, traveling waves occur within clusters but do not propagate continuously between clusters.

Our interhemispheric cross-correlation measure captures many of these dynamics (Fig. 8b). For example, it is sensitive to metastable transitions in partially synchronized waves, as evidenced by periods of strong interhemispheric cross-correlation punctuated by brief desynchronizations (Fig. 8c). This measure is also sensitive to metastable dynamics even in cases where the FC is globally weak (Fig. 8d). Conversely, there exist desynchronized states with strong FC but no temporal dynamics in the interhemispheric cross-correlation (e.g., $c = 0.6$, $\tau = 2$ ms). Lurching waves exhibit two time scales, with relatively slowly evolving correlation structure interleaved with a faster cluster-switching time (Fig. 8e).

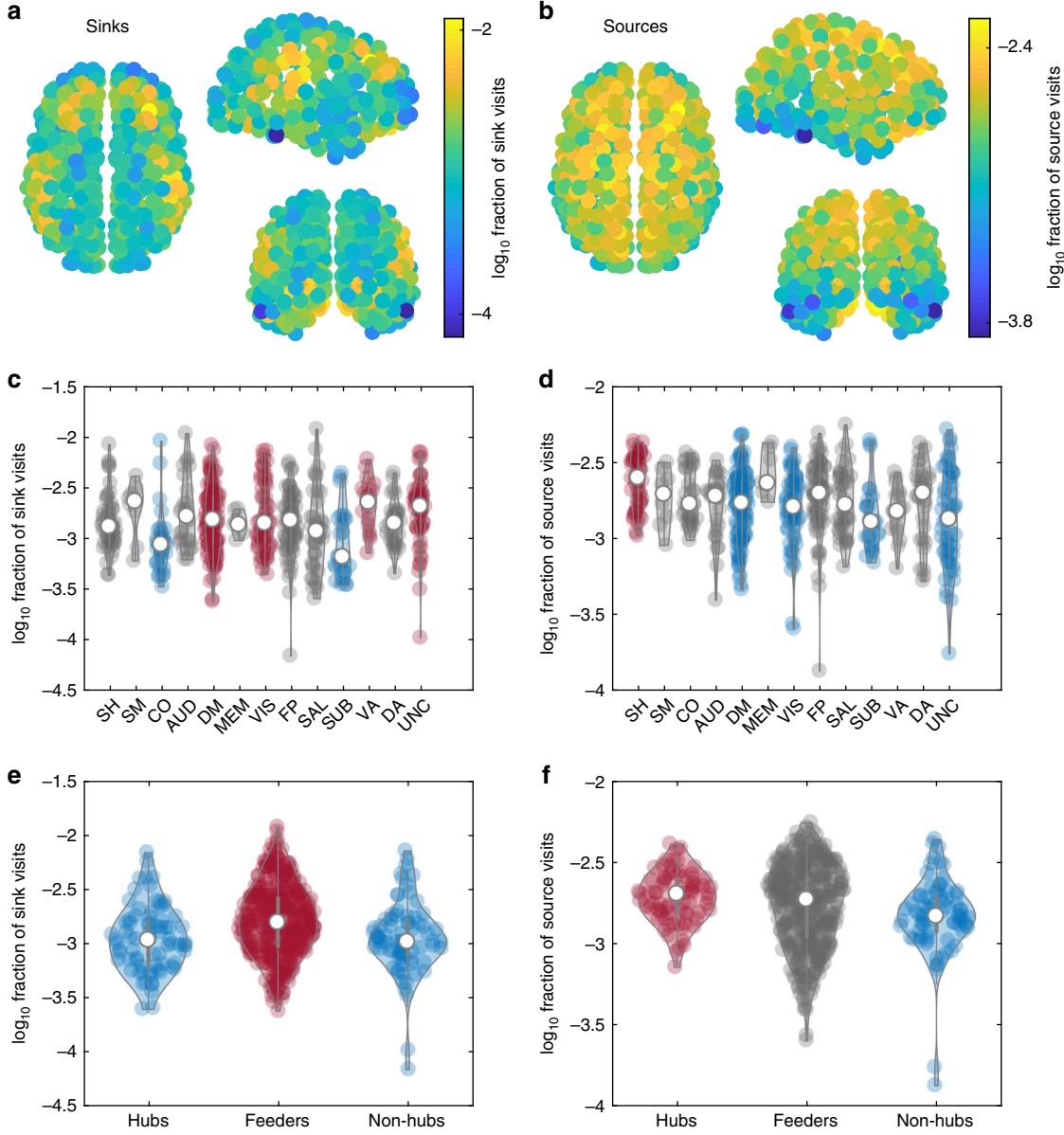

**Fig. 7** Sink and source properties. **a** Spatial distribution of sinks. **b** Spatial distribution of sources. **c** Overlap of sinks with functional networks. Blue denotes networks with fewer visits than red and gray denotes no significant difference from any other group. Networks are labeled as follows: AUD auditory, CO cingulo-opercular, DA dorsal attention, DM default mode, FP fronto-parietal, MEM memory, SAL salience, SH somatomotor hand, SM somatomotor mouth, SUB subcortical, UNC unclassified, VA ventral attention, VIS visual. **d** Overlap of sources with functional networks. Colors as per **c**. **e** Overlap of sinks with hubs (top 75 nodes by strength), feeders, and non-hubs (bottom 75 nodes by strength). Colors as per **c**. **f** Overlap of sources with hubs, feeders, and non-hubs. Colors as per **c**. White circles in violin plots denote group medians; violins are kernel density estimates. Statistics for **c**, **d** are given in Supplementary Table 1 and for **e**, **f** in Supplementary Table 2. Source data are provided as a Source Data file.

Prior studies modeling neuronal dynamics on empirical connectomes have used the match between predicted and empirical FC to tune underlying parameters accordingly[54–57]. We hence tested how spatial waves compared with these other candidate spatiotemporal patterns in their match to empirical resting-state FC (Fig. 9). Intriguingly, we find that the smooth waves observed for the parameter combination of $c = 0.6$ and $\tau = 1$ ms yield the second highest correlation across the entire parameter space tested (with or without GSR), only marginally lower than the best global fit (for $c = 0.2$ and $\tau = 0$ ms), which exhibited large-scale waves coexisting with discrete clusters. Additional tuning of the model parameters could improve this fit. Moreover, triangulating model fit with other dynamic metrics— wave properties, dwell times, source/sink distributions—would

improve the identifiability of the model parameters from empirical data.

We also verified that the existence of wave patterns reproduces across connectomic data (Supplementary Text, Supplementary Movie 9). Waves similar to those in our fully connected weighted connectome exist also in sparser networks and occur whether using traditional weight-based thresholding or consistency-based thresholding[58]. Waves exist also on a 10% density binary network thresholded by weight. We also observed waves arising on two entirely independent connectomes: an elderly connectome derived using probabilistic tractography[59] and the 998-node Hagmann et al.[60] connectome derived using deterministic tractography from diffusion spectrum imaging. Notably, waves arise on the connectomes from individual subjects, with similar

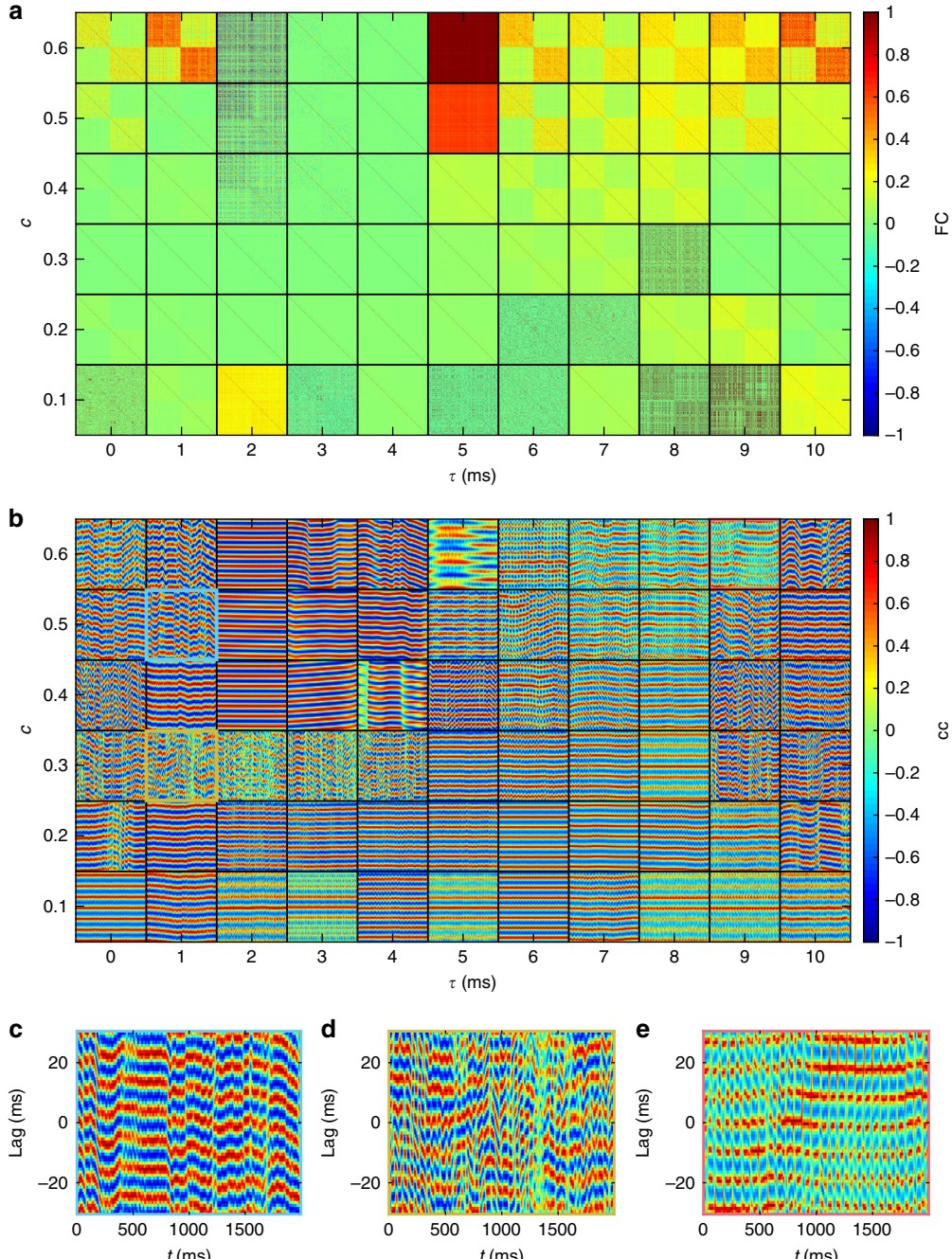

**Fig. 8** Dynamics as a function of coupling strength $c$ and delay $\tau$. **a** Functional connectivity matrices calculated directly from the neuronal time series (corresponding results after convolution of the neuronal time series with a hemodynamic response function are provided in Supplementary Fig. 4). Each tile shows one FC matrix with axes indexing the nodes 1–513. **b** Interhemispheric cross-correlation functions, showing a 1 s segment (time is on the horizontal axis in each tile) for lags between − 30 ms and 30 ms (lag is on the vertical axis in each tile). Exemplars in the text are highlighted here with colored outlines and shown in corresponding panels: **c** weakly coherent waves; **d** interhemispheric cross-correlation dynamics despite negligible average FC; and **e** lurching waves

speeds (subject mean ± SD = 33 ± 2 m s$^{-1}$) and similar dwell times (subject mean ± SD = 92 ± 8 ms) across the cohort (Supplementary Fig. 5a, b). Moreover, we verified robustness to initial conditions (Supplementary Text, Supplementary Fig. 5c, d).

To what extent does the structure of the human connectome contribute to the metastable wave dynamics? To address this question, we simulated the dynamics on a completely random network with the same number of nodes and same weight distribution as the original human connectome (the $R_{\mathrm{w}}$ surrogate of ref. [61]). To quantify the ensuing dynamics, we introduce local

and global synchrony order parameters $R_{\mathrm{local}}$, $R_{\mathrm{global}}$: the local but complex structure of waves causes a relatively high $R_{\mathrm{local}}$ but modest $R_{\mathrm{global}}$. For the combination of $c = 0.6$ and $\tau = 1$ ms the random network fully synchronizes, with $R_{\mathrm{local}} = R_{\mathrm{global}} = 1$ (Fig. 10a; see Methods for definitions of $R_{\mathrm{local}}$ and $R_{\mathrm{global}}$). Weaker coupling desynchronizes the nodes, but does not yield spatially coherent dynamics ($R_{\mathrm{local}}$ and $R_{\mathrm{global}}$ both low). That is, the purely random network does not generate waves, because the randomization scrambles the spatial relationships in the dynamics, rendering the spatiotemporal dynamics incoherent. It

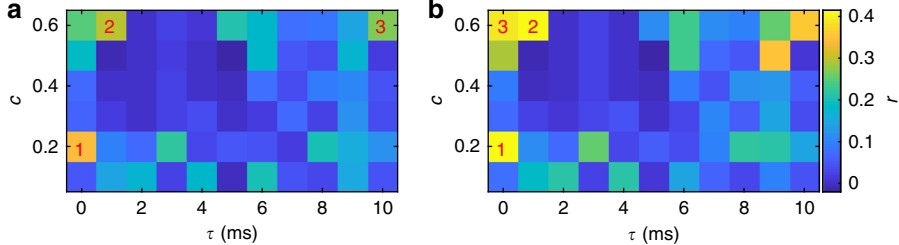

**Fig. 9** Correlation between modeled and empirical functional connectivity as a function of coupling and delay. **a** Pearson's correlation between empirical and modeled FC values for each pair of regions, without GSR. Numbers 1–3 indicate the top three highest correlations. **b** Same as **a** but with GSR

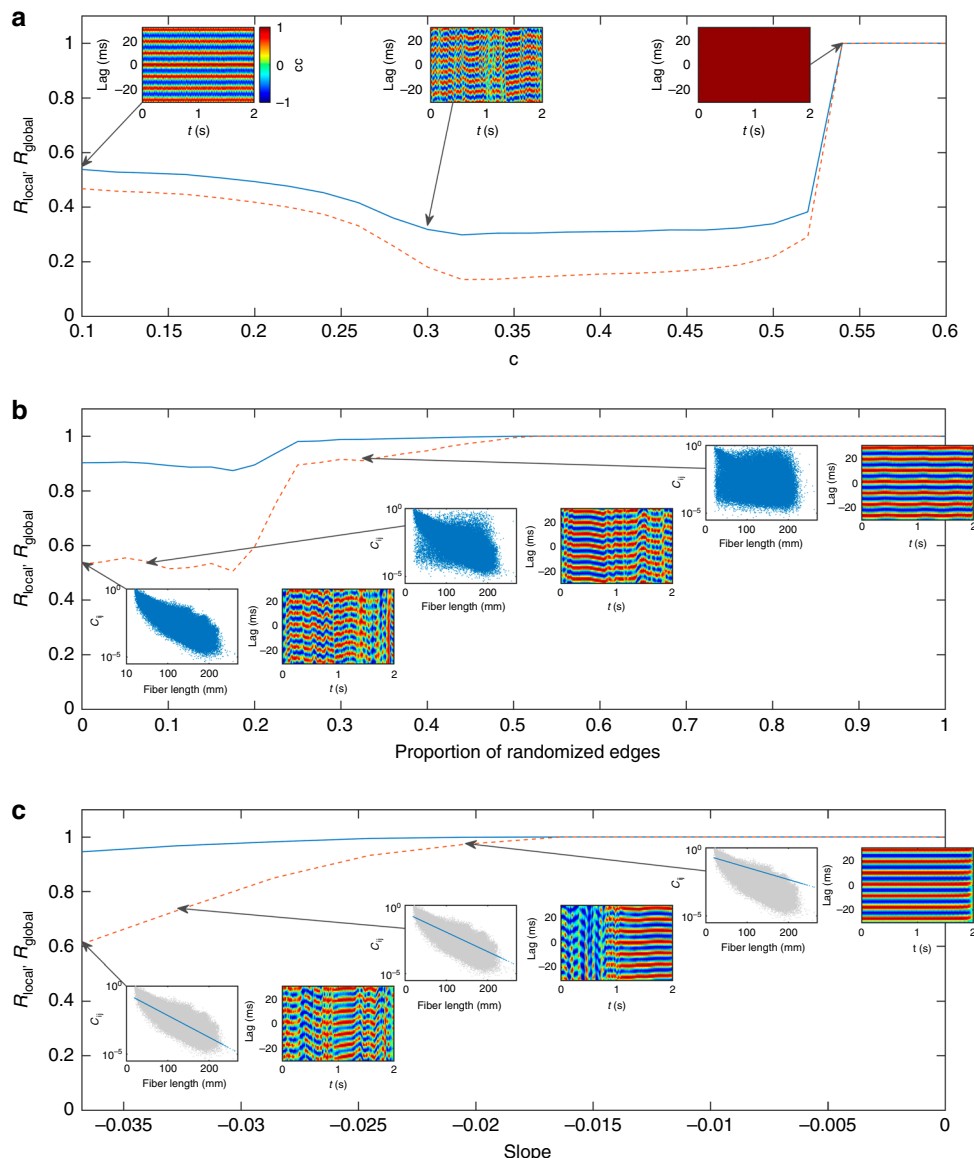

**Fig. 10** Local and global synchrony for surrogate networks. **a** Local and global synchrony in fully randomized networks as a function of coupling strength $c$. Lines show synchronization order parameters $R_{local}$ (solid) and $R_{global}$ (dashed). Insets show the interhemispheric cross-correlations at representative values of $c$, denoted by the arrows. **b** Loss of waves with progressive structural network randomization, parameterized by the proportion of randomized edges. Insets show the (log-)weight-vs.-fiber length relationships (blue point clouds) and the interhemispheric cross-correlations for the proportion of randomized edges denoted by the arrows. Lines are an average over an ensemble of ten random surrogates at each point. **c** Waves in synthetic networks with pure exponential weight–distance relationship, parameterized by the slope of the linear $\log_{10}$ (weight)-vs.-fiber length relationship. Insets show the (log-)weight-vs.-fiber length relationships for the synthetic exponential networks (blue) and the original network (gray), and the interhemispheric cross-correlations for the slope values denoted by the arrows. Source data are provided as a Source Data file

is noteworthy that random networks can support metastable dynamics, as evidenced by the interhemispheric cross-correlations around $c = 0.3$ having temporal structure. Thus, metastable transitions do not require a spatial embedding, although waves do.

We next sought to determine what happens between the case of the empirical connectome and the fully randomized connectome, by incrementally randomizing connections[61]. This randomization progressively destroys the spatial embedding of the connectome and, in doing so, introduces strong long-range connections[47,61]. For relatively shallow randomization ($< 15\%$ of edges randomized), the dynamics continue to exhibit strong local synchrony ($R_{local} \sim 0.9$) and moderate global synchrony ($R_{global} \sim 0.5$), similar to the real connectome (Fig. 10b). Thus, the existence of waves is robust to modest randomization of the network edges. These wave patterns are also metastable, but their lifetimes slowly increase (Fig. 10b, insets). From 20% to 25% of edges randomized, there is a rapid transition to a near-synchronized state ($R_{local} \sim 1$, $R_{global} \sim 0.9$), which fully synchronizes beyond ~50% randomization. Thus, the long-range connections introduced by network randomization tend to make the dynamics more stable and synchronized, eventually abolishing waves as the depth of randomization increases from the real brain to the random surrogate network. Hence, waves require a sufficiently strong spatial effect to emerge and they are resilient to modest random network perturbations.

To further understand the principles of wave generation, we next generated synthetic networks with a purely exponential weight–distance relationship (Fig. 10c). We find that this pure exponential network generates metastable waves, with slightly higher synchrony than the real brain ($R_{local} = 0.95$, $R_{global} = 0.61$), and similar lifetimes (Fig. 10c, left inset). Indeed, waves are also found for geometric surrogates that preserve the weight–distance relationship but are otherwise random[47], exhibiting waves with $R_{local} = 0.92$ and $R_{global} = 0.60$. To see how the waves depend on this exponential relationship, we tested a range of slopes (i.e., characteristic lengths) between the empirical best fit and the all-to-all network with no spatial embedding. It is noteworthy that again this has the effect of progressively increasing the strength of long-range connections. Similar to the partial randomization case, we find that increasing the characteristic length of the network connectivity increases the likelihood of long dwell times and eventually causes the network to synchronize. These results suggest that the key feature of the human connectome for generating waves is its spatial embedding, such that having short-range connections stronger than long-range ones facilitates the spatially localized synchrony required for waves. However, the specific properties of the waves on empirical networks—their spatial distributions of sources and sinks etc.—depend upon the specific configuration of the empirical connectome and cortical geometry.

In addition, we verified that wave patterns persist in the presence of weak noise (Supplementary Text, Supplementary Fig. 6, Supplementary Movie 10). Finally, we verified that emergent waves are not restricted to our particular choice of neural mass model. We tested two additional models: a network extension of the Wilson–Cowan model[45,62] and the Kuramoto model[63,64] (see Methods for details). In both models we found large-scale waves (Supplementary Text, Supplementary Movie 11).

## Discussion
Although metastability and waves in neural systems have been studied in isolation, here we show for the first time that these dynamical regimes are compatible, can arise from the human connectome, and can be explained with a unified mechanism. We have developed analysis tools to quantify these dynamics: one sensitive to coarse-grained phase dynamics, one sensitive to the spatiotemporal details of the underlying dynamical flows, and one to detect recurring spatiotemporal patterns across transitions. These new methods (available in the public domain) enable tests of our model predictions and open new avenues in the analysis of large-scale brain activity.

The distribution of sources and sinks in the wave flows organizes the spatiotemporal structure of the dynamics. This suggests a different conceptualization of how information can flow around the connectome. In network studies, the prevailing paradigm is that brain regions communicate via the structural connectivity, through a composite of the direct and indirect pathways between the two regions of interest. The net effect of the many paths remains inadequately described in this picture. Our results provide an interpretation in terms of spatial pathways along which activity preferentially flows—and hence potentially also pathways of information flow, e.g., via communication through coherence[65]. This is a higher-level description than simply following the fiber pathways, instead describing the net effect of communication between brain regions as mediated through the organization of the large-scale wave patterns. Moreover, we observe that metastable transitions in our simulations arise through the collision, annihilation, and/or creation of new sources and sinks. These dynamics underlie the co-occurrence of waves (the nature and distribution of these nodes) and metastability (their collision), hence providing a unifying mechanism for their co-occurrence.

The decomposition of waves into their sources and sinks naturally yields directionality in the dynamics, providing information inaccessible to standard FC. For example, sources can be thought of as controlling the resulting wave pattern and, intriguingly, we find that these tend to associate with hub regions more so than non-hubs. This may also speak to the purported role of costly rich-club connections between hubs in coordinating dynamics[43]. Wave sources and sinks also overlapped heterogeneously with functional network properties. Although there is substantial overlap, this suggests that waves may have functional specificity in the resting state. For instance, one subnetwork in which sinks tend to congregate is the default mode, consistent with its purported unique role in task-free/resting-state acquisitions. Task constraints may lead to a reorganization of wave patterns with a consequential increase in the location and functional specificity of sources and sinks. These observations are consistent with the presence of waves of rodent cortex, as evident in VSD recordings, exhibiting heterogeneous spatial distributions including patterns of sources and sinks[31] and bilateral symmetries[32]. These patterns reorganize upon brief whisker stimulation and include recurring sensory motifs embedded in spontaneous activity[31]. Moreover, wave patterns overlap with patterns of long-range structural connectivity, suggesting a role for the connectome in shaping the dynamics. It is also worth noting that waves contain information in their wavelength and carrier frequency that can be read out by dendritic trees with the matching orientation and spatial wavelength[66]—the dynamic and metastable patterns could hence yield distinct cognitive and behavioral consequences. Moreover, the spontaneous transitions in metastability are similar to the itinerancy that emerges in coupled oscillator systems[67] and to the heteroclinic cycles that have been proposed as a mechanism for sequential cognitive processes and decision-making[68]. Such theories of functional switching could be tested by moving beyond dwell times to study transition probabilities between specific patterns, as has been done for 2D waves[12] and transient MEG patterns[1]. Future application of our modeling and analysis methods to task-related data will enable elucidation of these potential functional roles.

In the past decade, much of our understanding of large-scale resting-state brain dynamics has been obtained through the lens of fMRI. With the exception of slow hemodynamic waves in visual cortex[69], it is not expected that fast neuronal wave dynamics can be easily resolved using fMRI. Faster imaging technologies are needed, such as MEG, ECoG, and VSD. Indeed, excellent spatiotemporal resolution is the norm in the animal preparations where waves have been studied in the most detail. For human neuroimaging, progress will require source reconstruction techniques that appropriately accommodate the complex amplitude–phase relationships and lagged covariances that reflect metastable waves.

Most of the present analysis was performed in the absence of noise. Intrinsic sustained fluctuations in this system emerge from the chaotic node dynamics. However, we also showed that the observed metastable wave dynamics are robust to the addition of modest system noise. Incorporating noise and other input stimuli also opens the possibility of exploring multistability—i.e., extrinsically perturbed jumps between states[2,70]. The present model is capable of exhibiting multistability: e.g., for $c = 0.5$ and $\tau = 15$ ms; wave or discrete cluster dynamics may emerge, depending on the choice of initial condition.

A key parameter in our model is the coupling strength, which scales the overall influence of activity from connected regions. There are several important points here. First, the connectome exhibits a rapid, roughly exponential drop-off in connection weight with distance[47]. In fact, this exponential drop-off is similar to the connectivity kernels used in neural field models, where pattern formation is well-known[40]. Although there has been some debate in the field as to the relative merits of continuum neural field models vs. networks of discrete coupled neural masses, we contend that these two approaches are more complementary than previously appreciated, beyond the trivial point that continuum models are routinely discretized for numerical simulations. Second, given the exponential spatial kernel, most of the coupling influence is relatively local ($\lesssim 50$ mm). Local patches of this scale are often lumped together in typical coarse parcellations of 50–100 nodes. This simplification reduces the spatial resolution and may hinder wave formation, which is a spatiotemporal collective phenomenon of many regions. Third, the most strongly coherent waves were observed for strong coupling. It is possible that highly coherent waves observed here are more common in states such as seizures, anesthesia, and sleep[4] than in healthy awake adults. The locally less coherent and lurching waves we observed may be more likely candidates for the higher complexity associated with conscious states. The lurching waves encompass both the functional segregation offered by sequentially activated discrete clusters and the transient localized wave patterns observed empirically[8].

Another important component of the coupling between regions is the propagation delay. The best fits to FC and dwell times were observed for stronger coupling combined with either short delays ($\tau = 1$ ms) or longer delays occurring across relatively large regions of parameter space (e.g., for $c = 0.4$–0.6, $\tau = 6$–10 ms). These parameters fall within biologically realistic limits (with the stronger coupling and short delays emphasizing the role of nearby nodes). We made the simple approximation of uniform delays between all regions, where the delay is interpreted as a mean effective delay. This is a common approach and, although there is evidence for regional variation in myelination supporting near-uniform delays between thalamus and cortex[71], it remains an approximation. The next-simplest approximation would be to assume a fixed conduction velocity, implying a distribution of delays increasing with distance, as used in some studies[41,45]. The reality is that both delays and velocities vary between regions[72]. As distributed delays have been linked to increased stability of

dynamics[73], we conjecture that wave dynamics may be more stable to perturbation with broader delay distributions.

We also showed that waves are not specific to the neural mass model we employed but also arise on the human connectome when using the Wilson–Cowan model[45,62] and the Kuramoto model[63,64]. These models fundamentally differ in the nature and time scales of their internal dynamics, the degree of mathematical and physiological abstraction, and the coupling mechanisms. Prior studies of the Kuramoto model have shown that time delays and spatially constrained connectivity can engender the sort of multi-frequency effects presently observed[55,56,64]. In addition, although not formally analyzed, a recent study of brain eigenmodes using the Wilson–Cowan model also supports the emergence of traveling waves[36]. The presence of waves across all three models speaks to principles that are independent of particular choices of models or their parameters, although future quantitative analyses are required to understand how these particularities influence the time scales and other properties of the ensuing waves.

The metastable lifetimes depend on the synchronizability of the network (Fig. 10), the neural mass membrane time scales, and inter-node delays. Transitions result from the tension between different regions attempting to synchronize their phases with their locally connected neighbors, with each transition (and hence lifetime) precipitating from this tension giving way to a new dynamically evolving phase-synchrony pattern. The situation is analogous to chimera states and metastability in other nonlinear dynamical systems coupled in a spatially dependent manner[39,64]. We observe that the nature (probability distribution) of the metastable transitions of our model bear close resemblance to those derived from the analysis of metastable switching in resting-state MEG data[1,52], although there is a modest quantitative mismatch in the precise form. Reducing this mismatch could act as a cost function if fitting the wave dynamics more formally to empirical data. Both sets of dwell times are broadly similar to a lognormal distribution, although formal testing shows other long-tailed distributions (e.g., exponentially truncated power law) can provide a similar quality fit to the upper tails. Alternatively, the hidden Markov model method employed in the MEG study could be applied to our simulations, after application of an appropriate forward model. There is also scope to improve our phase-flow streamline methods.

In sum, waves of cortical activity appear in empirical recordings, across broad recording modalities, species, and behavioral states. Here we show that the human connectome supports the spontaneous emergence of complex metastable brain waves. These are robust to changes in model parameters, the choice of model, and the thresholding of the underlying connectome. They reproduce across connectome datasets and yield qualitative matches with empirical resting-state data. The widespread empirical observation of traveling wave phenomena reproduced suggests a range of functional and potentially pathological roles in cognition and states of awareness[15]. The present modeling study makes specific predictions that can be further tested in these data and, moreover, yields novel quantitative analysis techniques that allow for a fundamental shift from static to non-stationary analytic frames.

## Methods

**Connectomic data.** We derived estimates of whole-brain structural connectivity from diffusion images of 75 healthy subjects (aged 17–30 years, 47 females). Diffusion MRI data were acquired on a Philips 3 T Achieva Quasar Dual MRI scanner (Philips Medical Systems, Best, Netherlands). We estimated the fiber orientation distribution (FOD) within each voxel constrained spherical deconvolution, implemented in MRtrix[74]. Tractograms were generated using a probabilistic streamline algorithm[74], which produces a set of connection trajectories by randomly sampling from the orientation uncertainty inherent in each FOD along the

streamline paths. Our connectivity matrices were reconstructed from densely seeded tractography ($10^8$ seeds) and parcellated into a relatively fine representation of 513 uniformly sized cortical and subcortical regions[75]. The resulting weighted, undirected matrices were nearly fully connected in each subject. The weights are the number of streamlines linking each pair of regions, divided by the streamline lengths (see Ref. [47] for full details). Results refer to a group-average connectome unless otherwise indicated.

Resting-state fMRI was recorded from the same subject cohort; 69 subjects had a full set of both diffusion and resting-state images. Functional images were collected using a T2* weighted echo-planar imaging sequence (188 images, echo time = 30 ms, repetition time = 2000 ms, flip angle = 90°, field of view 250 mm, 136 × 136 mm matrix size in Fourier space) and consisted of 29 contiguous 4.5 mm axial slices (no gap) covering the entire brain. Participants were requested to clear their mind without falling asleep. Preprocessing of data used included realignment, unwarping, anatomical co-registration, and spatial normalization. The functional data were corrected for white matter and cerebrospinal fluid signal. Further details of image acquisition and preprocessing are provided in ref. [76].

These structural and functional data were analyzed following approval from the QIMR Berghofer Human Research Ethics Committee (HRECp1476). Written informed consent was obtained from all participants following local institutional ethics approval.

**Neural mass model.** The neural mass model we used has been presented in detail elsewhere[46,70]; here we give a brief overview. This simple conductance-based neural mass model has three state variables at each node $j$: mean membrane potential of local pyramidal cells $V_j$, mean membrane potential of inhibitory interneurons $Z_j$, and the average number of open potassium ion channels $W_j$. Its dynamics are governed by:

$$\frac{dV_j}{dt} = -\left\{ g_{Ca} + r_{NMDA} a_{ee} \left[ (1-c)Q_V\left(V_j\right) + cQ_j^{network} \right] \right\} m_{Ca}\left(V_j\right)\left(V_j - V_{Ca}\right)$$
$$- \left\{ g_{Na} m_{Na}\left(V_j\right) + a_{ee}\left[ (1-c)Q_V\left(V_j\right) + cQ_j^{network} \right] \right\}\left(V_j - V_{Na}\right)$$
$$- g_K W_j\left(V_j - V_K\right) - g_L\left(V_j - V_L\right)$$
$$+ a_{ie} Z_j Q_Z\left(Z_j\right) + a_{ne} I_0, \tag{1}$$

$$\frac{dZ_j}{dt} = b\left[ a_{ni} I_0 + a_{ei} V_j Q_V\left(V_j\right) \right], \tag{2}$$

$$\frac{dW_j}{dt} = \phi\left[ m_K\left(V_j\right) - W_j \right]. \tag{3}$$

Here, delayed inputs from other regions in the network enter through the term $cQ_j^{network} = c \sum_k C_{jk} Q_V\left(V_k(t-\tau)\right) / \sum_k C_{jk}$, where $\tau$ is the delay time, $c$ is the global coupling strength, and $C_{jk}$ is the connectivity weight from region $k$ to region $j$. In Eq. 1, inhibitory input is a function of inhibitory activity and inhibitory membrane potential. Eqs 1–3 are non-dimensionalized to have unit capacitance such that non-dimensional time is numerically equivalent to milliseconds[46]. Here, $I_0$ is non-specific input to excitatory and inhibitory populations. The model distinguishes between AMPA and NMDA channels, where $r_{NMDA}$ denotes the ratio of NMDA receptors to AMPA receptors, and $a_{xy}$ terms parameterize the strength synaptic coupling from population $x$ ( $= e,i,n$, where $e$ and $i$ are the excitatory and inhibitory populations, respectively, and $n$ is a nonspecific input) to population $y$ ( $= e,i$). Parameters $b$ and $\phi$ are rate parameters (inverse time constants) that determine the time scales of $Z$ and $W$, respectively. The $g_{ion}$ terms in Eq. 1 are conductances of the corresponding ion channels and the $m_{ion}(V)$ functions describe the voltage-dependent fractions of open channels. They take the sigmoidal form

$$m_{ion}(V) = 0.5\left[ 1 + \tanh\left(\frac{V - T_{ion}}{\delta_{ion}}\right) \right], \tag{4}$$

where $T_{ion}$ and $\delta_{ion}$ are the mean and SD, respectively, of the threshold membrane potential for a given ion channel. The self-feedback of the pyramidal cells is split into a conventional voltage-dependent term for sodium channels $a_{ee}Q_V(V)(V - V_{Na})$ and a state-dependent term for NMDA-gated calcium channels, $r_{NMDA}a_{ee}Q_V(V)m_{Ca}(V)(V - V_{Ca})$.

The voltage-dependent functions $Q_V$ and $Q_Z$ are the mean firing rates of the excitatory and inhibitory populations, respectively, also given by sigmoidal forms

$$Q_V(V) = 0.5 Q_{V_{max}}\left[ 1 + \tanh\left(\frac{V - V_T}{\delta_V}\right) \right], \tag{5}$$

$$Q_Z(Z) = 0.5 Q_{Z_{max}}\left[ 1 + \tanh\left(\frac{Z - Z_T}{\delta_Z}\right) \right], \tag{6}$$

where $Q_{V_{max}}$ and $Q_{Z_{max}}$ are the maximum firing rates of the excitatory and inhibitory populations, respectively, and $V_T$ and $Z_T$ are the corresponding thresholds for action potential generation, and $\delta_V$ and $\delta_Z$ are the standard deviations in these thresholds.

The inhibitory population $Z$ is passively slaved to the pyramidal population: its membrane potential (and resulting firing rate) is driven by the firing rate of the output of the pyramidal cells, responding on a slow time scale parameterized by the factor $b$. The inhibitory population thus acts as a passive low-pass filter of the pyramidal cells.

Noise simulations were performed by replacing the input current term $a_{ne}I_0$ in Eq. 1 with $a_{ne}[I_0 + \sigma\eta(t)]$, where $\sigma\eta(t)$ is zero-mean Gaussian white noise with SD $\sigma$[70]. The model was solved for $\tau = 0$ using the Heun scheme with a time step of 0.01 ms.

Parameter values are given in Supplementary Table 3.

**Functional connectivity.** To estimate FC from the model simulations, we calculated the pairwise linear Pearson's correlation coefficient between each pair of time series. We calculated FC both on the mean pyramidal cell body potential (Fig. 8a), and after estimating the blood oxygen level dependent (BOLD) signal (Supplementary Fig. 4). We approximated the BOLD signal for each region using linear convolution with a hemodynamic response function (as implemented in SPM12's spm_hrf). This entailed downsampling to a 0.1 s time step using an antialiasing Chebyshev Type I IIR filter of order 8 (as implemented in MATLAB's decimate function). We compared this convolution method with full solution of the non-linear Balloon–Windkessel model[77] and found the results similar, so employed the faster convolution method.

To compare the model FC with FC derived from MEG data, we used publicly available amplitude–envelope correlations in ref. [45]. We used the same approach as ref. [45] to derive the amplitude–envelope correlations from the model: briefly, we mapped our 513-node parcellation onto the 68-node parcellation used in the MEG analysis by taking a weighted average within each coarse parcel (weighted by overlap). We then bandpass filtered to 8–13 Hz, orthogonalized the signals, calculated the Hilbert amplitude envelopes, downsampled to 1 Hz, and calculated Pearson's correlations between all pairs of amplitude time series. We then compared our model FC with the group-average FC (across 55 subjects) in the MEG dataset.

**Hidden Markov model.** The transition times for the Hidden Markov model were as derived in ref. [52]. In general, the number of states determines the level of detail in the analysis, such that increasing the number of states causes states to split, yielding a hierarchical view of the data. Twelve was chosen without any claim that this is the biological truth, but with checks for reliability of the state assignments across half-splits in the data. Alternative approaches include use of free energy for model selection purposes in a manner that balances model likelihood with model complexity.

**Interhemispheric synchrony.** We use a measure of interhemispheric synchrony derived as follows. Using the Hilbert phase $\phi_j(t)$ at each node $j$, we calculate the coherence for a set of nodes $S$ in terms of an order parameter $R_S(t)$, given by

$$R_S(t) = \frac{1}{|S|}\left| \sum_{j \in S} e^{i\phi_j(t)} \right|. \tag{7}$$

We can then calculate the two intrahemispheric coherences $R_L(t)$ for nodes $L$ in the left hemisphere and $R_R(t)$ for nodes $R$ in the right hemisphere. We define the interhemispheric synchrony as the sliding-window time-lagged cross-correlation $C(t,l)$ between $R_L(t)$ and $R_R(t)$. This quantity depends on time (via the windowing) and the cross-correlation lag $l$. We use windows of length 100 ms and 90% overlap; similar to all sliding-window methods, there is a tradeoff between resolution and uncertainty in the estimates within each window. Here we opt for increased smoothing at the expense of temporal resolution.

Time series of the time-lagged cross-correlation were thresholded to identify transitions between different spatiotemporal patterns. Transitions correspond to low cross-correlation for all time lags. We calculated $1/\text{Var}[C(t,l)]$, the inverse variance across lags $l$ at each $t$. We then thresholded this quantity by calculating its mean and finding all suprathreshold time intervals. Transition times correspond to the peak in $1/\text{Var}[C(t,l)]$ within each suprathreshold interval. The dwell-time distributions are robust to modest changes in the value of this threshold.

**Local and global synchrony**. We used the order parameter of Eq. 7 to calculate global synchrony $R_{global} = \langle R_{network}(t) \rangle_t$, and local synchrony $R_{local} = \left\langle \left\langle R_{L_m}(t) \right\rangle_m \right\rangle_t$, where $\langle \cdot \rangle_t$ and $\langle \cdot \rangle_m$ denote averages over time and nodes, respectively. Here, $R_{network}(t)$ is the coherence for the whole network, whereas $R_{L_m}(t)$ is the coherence for the set of nodes $L_m$ within a radius of 20 mm of node $m$ (including itself). For our parcellation and this radius, the local neighborhoods contain 11 nodes on average (range 3–17). $R_{local}$ is sensitive to local coherence and is hence high for waves. $R_{global}$ is high for globally synchronized states and takes low-to-moderate values when waves are present.

**Wave velocities**. We calculated the velocity vector field at each time point using a method similar to that of Rubino et al.[9]. First we extracted the instantaneous phase at each node using the Hilbert transform. The velocity **v** can be calculated from the spatial and temporal derivatives of the phase $\phi(x,y,z,t)$, as $\mathbf{v} = -\left( \left|\frac{\partial \phi}{\partial t}\right| / \|\nabla \phi\|^2 \right) \nabla \phi$. To calculate the spatial derivatives, we used the constrained natural element method[78], a meshless method for solving calculus problems on non-convex domains. This allowed us to calculate nodal quantities (specifically the components of the gradient vector) without needing to interpolate to and from a three-dimensional grid, which can introduce edge effects for nodes on the brain's convex hull. To handle phase unwrapping, we calculated the gradient of $e^{i\phi(x,y,z,t)}$ and used the identity $\frac{\partial \phi}{\partial x} = -ie^{-i\phi} \frac{\partial}{\partial x} e^{i\phi}$.

**Flow streamlines**. To calculate streamlines following the velocity vector field, we used a simple Euler stepping routine starting from the individual nodes. Flow at each point was estimated using constrained natural neighbor interpolation[78]. We used a maximum step length of 8 mm and stopping criteria of the streamline leaving the brain or reaching a maximum of 200 steps. Dense clusters of streamline points were identified by first discarding streamlines shorter than 20 steps (these tended to exit the brain rather than converge onto a source/sink), then discarding transients by restricting attention to the last 5 points of each streamline, and passing these to a density-based clustering algorithm DBSCAN[79]. Parameters of the clustering algorithm were a cluster radius of 6 mm and a minimum cluster size of 10 points.

**Replication datasets and models**. We verified our results using independently acquired connectomic data from both the elderly connectome[59] and the higher spatial resolution 998-node Hagmann et al.[60] dataset. We also exhibited waves in two different models: the Wilson–Cowan model extended in network form[62] and the Kuramoto model[63,64].

In the Wilson–Cowan network model, each region $j$ is described by two state variables $u_j$ and $v_j$, representing firing rates for excitatory and inhibitory populations, respectively. Their dynamics are governed by

$$\frac{du_j}{dt} = -u_j + f\left( a_{ee}u_j - a_{ie}v_j - z_e + cu_j^{network} \right), \tag{8}$$

$$\tau_0 \frac{dv_j}{dt} = -v_j + f\left( a_{ei}u_j - a_{ii}v_j - z_i \right), \tag{9}$$

where $f(x) = 1/[1 + \exp(-x)]$ is a sigmoidal firing rate function and delayed inputs from other nodes in the network enter via $u_j^{network} = \sum_k C_{jk}u_k(t - \tau)/\sum_k C_{jk}$, where $C_{jk}$ is the connectivity weight from region $k$ to region $j$. The simulations here used[62] $a_{ee} = a_{ie} = a_{ei} = 10$, $a_{ii} = -2$, $z_e = 1.5$, $z_i = 6$, $\tau_0 = 1$, $c = 5$, and $\tau = 2$.

In the Kuramoto model, each region is described by a single-state variable $\theta_j$ describing a local oscillatory phase and a natural frequency $\omega_j$. Their dynamics are governed by

$$\frac{d\theta_j}{dt} = \omega_j + c \sum_k C_{jk} \sin\left( \theta_k - \theta_j \right), \tag{10}$$

where $c = 0.0028$ is a constant set in previous work to match empirical FC results[63]. In this example, there are no delays between regions. Natural frequencies $\omega_j$ were determined as a function of the anatomical node strength $s_j = \sum_{k=1}^{513} C_{jk}$ to incorporate a hierarchy of time scales across the cortex. As in previous work[63], each region's intrinsic frequency is given by

$$\omega_j = a - (a - b)\left( \frac{s_j - s_a}{s_b - s_a} \right)^2, \tag{11}$$

where $a = 0.1$ Hz and $b = 0.01$ Hz are the maximum and minimum oscillatory frequencies, and $s_a = \min(s)$ and $s_b = \max(s)$ are the corresponding maximum and minimum strengths. It is noteworthy that the time scales of this proof-of-principle use of the Kuramoto model are tuned to match those of the slow BOLD response—as with other recent studies[54]—and hence differ from prior objectives to directly match the faster frequencies of MEG data[56]. As such, the parameterizations

(coupling strength) are not directly comparable, although we note that these dynamics are in a regime where the Kuramoto order parameter has mean 0.78 and SD 0.14 across the time series. It is worth noting that we do not include phase delays (equivalent to time delays) in these illustrative simulations. Prior work suggests that time delays may include additional dynamic instabilities[64] and intermittent frequency slowing[55].

**Reporting summary**. Further information on experimental design is available in the Nature Research Reporting Summary linked to this article.

**Code availability**. MATLAB code is available at http://www.sng.org.au/Downloads

## Data availability
All data are available from the corresponding authors upon request. The source data underlying Figs. 2b-d, 4, 7, and 10 and Supplementary Figs. 1, 5, and 6a,c are provided as a Source Data file.

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

## Acknowledgements

This work was supported by the National Health and Medical Research Council (Project Grants 1145168 and 1144936, Program Grant 1037196, and Fellowships 1110975 and 1118153) and the Australian Research Council (Centre of Excellence for Integrative Brain

Function CE140100007). M.W.W. and R.A. are supported by the Wellcome Trust (203139/Z/16/Z and 106183/Z/14/Z) and the MRC UK MEG Partnership Grant (MR/K005464/1). MEG state dwell-time data come courtesy of data acquired by Ben Hunt and Matt Brookes from the University of Nottingham as part of the MRC UK MEG Partnership Grant.

## Author contributions

J.A.R., L.L.G., and M.B. designed the study. J.A.R. and L.L.G. performed the simulations. J.A.R., L.L.G., R.A., and M.B. performed the analyses. R.A. and M.W.W. provided the MEG resting-state dwell times and FC matrices. P.B.M. and G.R. provided the structural and functional MRI data. All authors discussed the results and contributed to writing the paper.

## Additional information

**Competing interests:** The authors declare no competing interests.

