## [Peer Review File · Nature Communications]

Reviewers' Comments:

Reviewer #1:

Remarks to the Author:

In this paper, the authors discuss a model of the dynamics of whole brain activity. The model consists of a 3rd order differential equation at each node of a 531 node network with connectivity derived from human tractography. They simulate the model, and develop data analysis methods to characterize the resulting dynamics. They identify sets of space-time patterns, and abrupt transitions between these patterns. Novel aspects of the work include the development of tools to analyze these complex data, the proposal that activity evolves through transient configurations of sources and sinks, and the study of wave dynamics on complex structure network derived from human tractography.

Major comments

1. The simulations and data analysis are very nice. The paper does a nice job describing what is happening, but at times does not discuss deeply why certain features occur. Why is using a real connectome important? Is there something special about the human connectome? Or would a simulated connectome (random or small world) produce similar results? Does the structure determine the patterns that emerge? Why are the metastable states so brief (100 ms)? What precipitates the transitions? There are 75 subjects with tractography, are the dynamical patterns the same for all of them, or do differences matter? Once a pattern transitions, does it later reappear? I don't think it's necessary to answer all of these questions, and it's good that the paper inspires so many interesting questions. But I wish I understood more about the nature of these patterns and their transitions.
2. Maybe I missed this, but I did not understand how the results from the different subjects (75) are averaged. Were dynamics simulated on each patient's connectome? How were the results averaged? Were the simulations run multiple times with different initial conditions, and the results averaged across runs?
3. The simulated metastable dynamics are fast. The simulations are compared with slow fMRI results. Despite the "empirical evidence that short-lived states visible in electrophysiology do leave an imprint on the correlations between regions at time scales visible to fMRI", the model results may be better compared with the MEG data. Could other features like wavelength, speed, and FC be compared between the simulation and the MEG data, in addition to the dwell time reported in Figure 2? The sources and sinks in the model are relatively stable on time scales of ~100 ms, and does this also occur in MEG data? This would also avoid the need to convolve the simulated data with a hemodynamic response function.
4. I was at times confused by the "functional network" terminology. From the Abstract, one claim is that this work moves "the study of functional networks from a static to an inherently dynamic frame". I use the definition from Sporns to define a functional network as nodes with coupled activity, which requires a "dynamic frame" to observe. On page 13, the "classic functional networks" are defined differently.

Minor comments

"Yet with the exception of the pathological strongly-nonlinear dynamics in epileptic seizures [5, 6], theory and modeling have fallen behind the body of empirical results." For more specific references to waves in seizures see (Smith, E., Liou, J.-Y., Davis, T., Merricks, E., Kellis, S., Weiss, S., et al. (2016). The ictal wavefront is the spatiotemporal source of discharges during spontaneous human seizures. *Nature Communications*.) and (Martinet, L.-E., Fiddyment, G., Madsen, J. R., Eskandar, E. N., Truccolo, W., Eden, U. T., et al. (2017). Human seizures couple across spatial scales through travelling wave dynamics. *Nature Communications*, 8, 14896.)

pg 4, Why is the nodal speed slower than the wave propagation speed?

Figure 2, Are these results for one simulation? Or for many simulations with different initial conditions?

Figure 2, Can nodal speed be related to structure connections? Would a plot of nodal speed vs structure network degree indicate this? My naive guess is that speed is related to structure in some way, since that's the only feature that differs between the nodes.

pg 7, "calculate the instantaneous coherence", I think the measure used is the phase locking value (PLV).

pg 7, "At the time of a metastable transition, the large-scale wave pattern breaks up and disorganized short wavelengths dominate. Thus, the metastable wave signatures are separated by narrow periods of time with relatively low correlation between the hemispheres." This conclusion is not clear in Figure 3. How do we see short wavelength and low correlation? How were black lines chosen in Figure 3b?

pg 8, "Extracting all times corresponding to low synchronization thus yields an automatically-computed set of transition times", please define "low", my apologies if I missed this.

pg 10, "Our model thus provides a plausible mechanism for these metastable transitions." More accurate to state that the model captures one aspect of the metastable transitions, the dwell time.

pg 10, Is the r computed for one subject, or for all subjects? Is FC computed for the entire trace, or for different stable intervals of the dynamics? Is the FC similar because both model and rsfMRI have same SC?

Figure 5 caption, "colored by orientation in the plane)", is the orientation in 3 dimensions, not 2 dimensions?

pg 12, "Even though we have not imposed any temporal smoothness (e.g. as done in optical flow methods [15]), the streamlines are well-behaved across time, reflecting the stability and order of the underlying activity patterns." The instantaneous phase at each node is extracted using the Hilbert transform, which may impose temporal smoothness.

pg 12, "The first (Fig. 5b, left) shows two frontal left-hemisphere sources and a sink that merge and reconfigure to yield one left frontal source, while in the right hemisphere one frontal source and one frontal sink move closer together and a temporal sink emerges. The second example (Fig. 5b, right) exhibits complex frontal clusters of sources and sinks that split and move posteriorly, plus the emergence of an occipital sink." Difficult to see these claims in Figure 5.

pg 13, "calculating the standard deviation of visits to each node", Visits of what to each node? What is the standard deviation of visits, is it a standard deviation of counts? In 20 ms, seems like very short window to capture variability of spatial dynamics?

pg 14, "while sources visit the somatomotor hand network more often than others including the default mode.", That's not completely clear from the figure, the distributions appear to overlap quite a bit. Maybe the mean value of fraction of visits is different.

pg 14, "This analysis confirmed that the feeder nodes act as sinks significantly more often than hubs and non-hubs (Fig. 6e), while the hubs act as sources significantly more often than the non-hubs (Fig.

6f)." p values? Also the effect could be significant but the difference appears to be very small.

pg 18, The first paragraph suggests the results are robust to changes in network structure. But it's not quantitative. The movie does provide evidence that waves exist in different network structures. But it's not clear that the waves are "similar to those in our fully-connected weighted connectome". That'd require more quantitative analysis. The second paragraph suggests the results are consistent to changes in the neural mass model. This is an interesting and important observation, but the discussion is suggestive and the results not quantitative.

pg 19, "While metastability and waves in neural systems have been studied in isolation, here we show for the first time that these dynamical regimes are compatible, can arise from the human connectome, and can be explained with a unified mechanism." Please review the unifying mechanism in the Discussion.

pg 19, "This suggests that waves have functional specificity." This is not clear from Figure 6c,d.

pg 19, "one subnetwork in which sinks tend to congregate is the default mode, consistent with its purported unique role in task-free/resting state acquisitions." this is very difficult to see in Figure 6c, the DM result does not stand out.

pg 23, Equation 1, Please verify inhibitory input is a function of inhibitory activity and inhibitory membrane potential?

pg 24, "are the corresponding thresholds for axon potential", action potential.

pg 26, "We define the interhemispheric synchrony as the sliding-window time-lagged cross-correlation $C(t,l)$ between $RL(t)$ and $RR(t)$." This measure is the cross-correlation of the phase locking values PLV. For "interhemispheric synchrony", why not compute PLV between brain regions across hemispheres?

Reviewer #2:

Remarks to the Author:

In this beautiful paper, the authors investigate a whole-brain model that exhibits propagating wave patterns similar to experimental findings. The authors show, by simulating resting state conditions, that different wave patterns are seen in succession, and they form metastable states (different from attractor states). The paper also proposes ways to reconcile different observations, and opens many avenues for further research, so it is a very useful contribution to the field.

I found that three aspects should be improved:

1. The authors should relate their work to that of Tim Murphy's lab (Mohajerani et al., Nature Neurosci 2013) where they found that motifs of activity (sometimes propagating) relate to axonal projections in mouse. They also found that propagating waves are all over the brain and many times symmetric between hemispheres (Mohajerani et al. J Neurosci 2010). Since some of these findings are similar to what is reported here, it would make sense to discuss the similarities and differences.
2. As the authors acknowledge, there is no noise considered, which seems to be justified by the fact that the "resting state" is supposed not to be influenced by external inputs. However, this would be true for deep slow-wave sleep, but I do not think one can consider the resting state as totally isolated. The authors should include a simulation, even very preliminary, in the presence of weak noise, to

show how the propagating patterns are affected. This is particularly important for transient (metastable) states, which could be very sensitive to noise. Is it the case here ?

3. The findings are here reported using only one particular model, which is a neural mass model introduced by the authors. To what respect the findings depend on this model, or would be general, do you have any evidence for this? (like a previous paper where the model was compared to another one?)

Minor:

On page 20, when you say that high time resolution is used, citing only MEG and ECOG, you should also include VSD, because many traveling wave studies have been using this technique.

Reviewer #3:

Remarks to the Author:

In this work the authors show the emergence of various kinds of wave dynamics in the brain at macro-scale using a neural mass model applied to the structural connectivity of the brain. Quite remarkably, the model dynamics not only show various types of waves, e.g. travelling, rotating, spirals, sources and sinks but also exhibit spontaneous transitions between these waves, even in the absence of noise. Such transitions provide further evidence for metastable dynamics, winnerless competition between different states or attractors, which is now increasingly often hypothesized and shown to underlie the brain dynamics. The metastable transitions observed in the numerical simulations of the model utilized here occur via brief desynchronisation periods during which wave patterns reconfigure, which the authors automatically characterize via interhemispheric cross-correlation.

Overall, I quite enjoyed reading this manuscript and I think the demonstration of the emergence of and metastable transitions between various wave dynamics, which have already been observed in empirical MEG data, is an important contribution. Below are some comments and questions that can improve the clarity of presentation of this manuscript and help better relate it to the existing models in the literature.

Major comments:

1. I think my main major comment is about tuning the model parameters to fit the empirical data. As one of their key contributions, the authors state on page 4 "We explore a range of coupling strengths and delays beyond the narrow area of parameter space previously studied". My main question is whether that parameter space is biologically realistic. Generally most computational models are applied by optimizing the model parameters to best fit the observed functional connectivity or some other characteristic of the fMRI or MEG data (e.g. Deco & Kringelbach 2014 *Neuron*, Cabral 2011 *Neuroimage*, Cabral 2014 *Neuroimage*, Deco 2017 *Neuroimage*). I find the wide range evaluation of the parameter space that the authors present, quite important in terms of understanding different dynamics the model is capable of creating, but I think it is also important to find out, which of the simulated dynamics are biologically meaningful, e.g. they exhibit similar characteristics as the empirical data in fMRI, MEG or functional connectivity matrices. For instance on page 17 Figure 7a, some of the presented functional connectivity matrices seem to be quite different than that of the empirical data. Can the authors please clarify this?

2. The paper generally presents a thorough literature study, but I think the following references are

also very relevant for this study: (Cabral 2014 Neuroimage) and (Deco 2017 Neuroimage), where the authors model MEG data using a Kuramoto and a multi-frequency Hopf model, respectively. Also, in reference 35, the authors also utilize a Wilson-Cowan model, and their videos show rotating as well as travelling waves. Although it has not been explicitly discussed as travelling wave dynamics, I think their supplementary videos and numerical simulations of Wilson-Cowan equations support the findings of this manuscript quite nicely. The authors may want to refer to these findings using the Wilson-Cowan model, in relation to their findings with this model, e.g. on page 18 last paragraph before Discussion.

3. The states in the MEG data are characterized using a hidden markov model (HMM) with 12 states (page 9). How is the number of states chosen, how would it affect the distribution to change the number of states in the HMM?

Minor Comments:

4. Page 5: Can the authors comment or speculate on what may be the reason that the travelling wave speeds increase with distance from midline (Page 5 and Fig. 2)?

5. Page 5: "The regions supporting faster wave fronts from a roughly contiguous zone in each hemisphere, spanning from the frontal lobe to the occipital lobe, via the parietal lobe and posterior aspects of the temporal lobe." How to these regions relate to sensory or higher level cortices?

6. How are the low synchronization time points identified; e.g. using a certain threshold, if so what is the threshold? For instance in Fig. 3 caption, the authors state: "(b) Vertical lines depict instances of low values of the interhemispheric cross-correlation function, corresponding to wave transitions". What threshold is considered to be low?

7. Fig 5: It would be clearer if the authors state that x represents the distance to the midline in Fig 5c, in the figure caption.

8. Can the authors please comment on why the sources and the sinks have the tendency to remain within a single hemisphere (page 13, end of first paragraph)? Would that be due to the relatively smaller number of interhemispheric connections in DTI?

9. Page 14: "This analysis confirmed that the feeder nodes act as sinks significantly more often than hubs and non-hubs (Fig 6e), while the hubs act as sources more often than the non-hubs". Can you please report the p-values or refer to Supplementary Table S2, if that is the corresponding p-values?

10. Page 16: "Waves are also primarily observed for stronger coupling with longer delays, occurring across a relatively large regions of parameter space (e.g. for $c=0.4-0.6$, $\tau=6-10ms$ ". Again, would this range agree with biologically meaningful parameter values; e.g. if the model were to fit to best represent the functional connectivity of the empirical data, would the estimated parameters fall into that range?

11. Page 18: "In the Kuramoto case we observed waves when the model is in a regime of partial synchronization". How does this range relate to the parameter range estimated by fitting the Kuramoto model to the resting state data; e.g. Cabral 2011 Neuroimage and Cabral 2014 Neuroimage, where the authors fit a Kuramoto model with delays to fMRI and MEG data?

12. Page 23: Please specify what c and C denote in Eqs 1,2 and what e and I denote in $x(=e,i,n)$.

13. Page 26: Methods, Functional connectivity: "We calculated functional connectivity both on mean pyramidal cell body potential, and after estimating the BOLD ": which version of the functional connectivity was then used for further analysis. I thought only the one after the convolution with the hemodynamic response was used, but that is not clear in the statement above. Can you please clarify?

14. Inter-hemispheric synchrony: I wonder what effect the separation of two hemispheres has for this analysis; i.e. could this synchrony measure also be applied with a different partitioning of the brain; i.e. front-to-back, and still capture the transitions, or is there a specific information in the inter-hemispheric synchrony?

15. Page 27, last paragraph, please state that C refers to the connectivity matrix, ideally first time you introduce C in Eq 1.

16. The authors chose use the Kuramoto model without delays, although that has been reported to be a crucial component for realistic simulations (Cabral 2011 and 2014 Neuroimage). As Kuramoto model is not the main focus of this paper, I think it is fine if the authors chose to proceed without including the delays to Kuramoto model, but I would suggest at least commenting on how that may affect the wave dynamics.

17. Supplementary Fig. S1: It may help using a different colormap to better distinguish different resting state networks and improve the clarity of presentation.

18. Supplementary Tables S1 and S2: I may be misunderstanding this analysis but can the authors please clarify the details of the comparisons presented in the supplementary tables, t-tests between exactly what quantities. I believe I fail to understand how to read the tables. In my understanding you test the significance of the overlap between a functional network and sinks (or sources) emerging within that network. Shouldn't this result in one vector of size (1 x nr of resting state networks) for the sinks and one vector for the sources rather than a matrix. Can the authors please clarify?

Reviewer Responses

Reviewer #1

In this paper, the authors discuss a model of the dynamics of whole brain activity. The model consists of a 3rd order differential equation at each node of a 531 node network with connectivity derived from human tractography. They simulate the model, and develop data analysis methods to characterize the resulting dynamics. They identify sets of space-time patterns, and abrupt transitions between these patterns. Novel aspects of the work include the development of tools to analyze these complex data, the proposal that activity evolves through transient configurations of sources and sinks, and the study of wave dynamics on complex structure network derived from human tractography.

We thank the reviewer for this positive appraisal of our work and constructive feedback. Our detailed responses are given below, interleaved with the reviewer's comments (in blue).

Major comments

1. The simulations and data analysis are very nice. The paper does a nice job describing what is happening, but at times does not discuss deeply why certain features occur. Why is using a real connectome important? Is there something special about the human connectome? Or would a simulated connectome (random or small world) produce similar results? Does the structure determine the patterns that emerge? Why are the metastable states so brief (100 ms)? What precipitates the transitions? There are 75 subjects with tractography, are the dynamical patterns the same for all of them, or do differences matter? Once a pattern transitions, does it later reappear? I don't think it's necessary to answer all of these questions, and it's good that the paper inspires so many interesting questions. But I wish I understood more about the nature of these patterns and their transitions.

These are all excellent questions and we are pleased that our work opens so many avenues, and indeed to address all these comprehensively would require multiple follow-up papers. Nevertheless we have performed several new analyses to address most of these points. Our responses to each question in turn:

Why is using a real connectome important? Is there something special about the human connectome? Or would a simulated connectome (random or small world) produce similar results?

To some extent, the real human connectome is special: we have added new analysis showing that a completely random network, with the same number of nodes and same weight distribution, does not yield waves. Instead, for the same coupling and delay parameters as the main results in the paper, the random network fully-synchronizes. Weaker coupling desynchronizes the nodes, but does not yield spatially-coherent wave dynamics. The reason for this is that the randomization scrambles the spatial relationships in the dynamics, rendering the local spatiotemporal dynamics incoherent and prohibiting wavelike patterns. Note random networks *can* support metastable dynamics, as evidenced by the interhemispheric cross-correlations around $c=0.3$ having temporal structure. Thus metastable transitions do not require a spatial embedding.

To quantify these relationships, we have added new analyses calculating two order parameters: mean global synchrony R_{global} across the whole network, and mean local synchrony R_{local} , measuring the coherence of a node with its neighbors within a local sphere, averaged across all nodes. Waves correspond to high local synchrony and low global synchrony. We show these results for a random network in the new Figure 10 and accompanying text on p21:

To what extent does the structure of the human connectome contribute to the metastable wave dynamics? To address this question, we simulated the dynamics on a completely random network with the same number of nodes and same weight distribution as the original human connectome (the R_w surrogate of Ref. [70]). To quantify the ensuing dynamics, we introduce local and global synchrony order parameters R_{local} , R_{global} . The local but complex structure of waves causes a relatively high R_{local} but modest R_{global} . For the combination of $c=0.6$ and $\tau = 1$ ms the random network fully synchronizes, with $R_{\text{local}}=R_{\text{global}}=1$ (Fig. 10a; see Methods for definitions of R_{local} and R_{global}). Weaker coupling desynchronizes the nodes, but does not yield spatially-coherent dynamics (R_{local} and R_{global} both low). That is, the purely random network does not generate waves, because the randomization scrambles the spatial relationships in the dynamics, rendering the spatiotemporal dynamics incoherent. Note that random networks can support metastable dynamics, as evidenced by the interhemispheric cross-correlations around $c=0.3$ having temporal structure. Thus metastable transitions do not require a spatial embedding although waves do.

Figure 10a: Local and global synchrony in fully-randomized networks as a function of coupling strength c . Lines show synchronization order parameters R_{local} (solid) and R_{global} (dashed). Insets show the interhemispheric cross correlations at representative values of c , denoted by the arrows.

We have added methods text for these synchrony measures in a new subsection on p33:

Local and global synchrony

We used the order parameter of Eq. (7) to calculate global synchrony $R_{\text{global}} = \langle R_{\text{network}}(t) \rangle_t$, and local synchrony $R_{\text{local}} = \langle \langle R_{L_m}(t) \rangle_m \rangle_t$, where $\langle \cdot \rangle_t$ and $\langle \cdot \rangle_m$ denote averages over time and nodes, respectively. Here, $R_{\text{network}}(t)$ is the coherence for the whole network, while $R_{L_m}(t)$ is the coherence for the set of nodes L_m within a radius of 20 mm of node m (including itself). For our parcellation and this radius, the local neighborhoods contain 11 nodes on average (range 3-17).

R_{local} is sensitive to local coherence and is hence high for waves. R_{global} is high for globally synchronized states and takes low to moderate values when waves are present.

To more deeply understand the role of the spatial properties of the connectome on the emergence of waves, we next examined what happens between the two extremes of the empirical brain and the fully randomized brain, by simulating the dynamics on progressively-randomized networks. This has the effect of progressively scrambling the spatial properties of the empirical network (0%) towards a fully random network (100%). Randomization tends to introduce long-range connections (Roberts et al. 2016, Gollo et al. 2018), similar to how small-world networks are constructed by partially rewiring regular lattices. We find that the long-range connections introduced by partial randomization induce a transition from wave dynamics to global synchrony. We have added a new Figure 10b and accompanying text on p23:

We next sought to determine what happens between the case of the empirical connectome and the fully-randomized connectome, by incrementally randomizing connections [70]. This randomization progressively destroys the spatial embedding of the connectome and in doing so introduces strong long-range connections [53, 70]. For relatively shallow randomization (<15% of edges randomized), the dynamics continue to exhibit strong local synchrony ($R_{local} \sim 0.9$) and moderate global synchrony ($R_{global} \sim 0.5$), similar to the real connectome (Fig. 10b). Thus the existence of waves is robust to modest randomization of the network edges. These wave patterns are also metastable, but their lifetimes slowly increase (Fig. 10b, insets). From 20 to 25% of edges randomized, there is a rapid transition to a near-synchronized state ($R_{local} \sim 1$, $R_{global} \sim 0.9$), which fully synchronizes beyond $\sim 50\%$ randomization. Thus, the long-range connections introduced by network randomization tend to make the dynamics more stable and synchronized, eventually abolishing waves as the depth of randomization increases from the real brain to the random surrogate network. Hence, waves require a sufficiently strong spatial effect to emerge, and they are resilient to modest random network perturbations.

Figure 10b: Loss of waves with progressive structural network randomization. Lines denote synchronization order parameters R_{local} (solid) and R_{global} (dashed) as a function of the proportion of randomized edges. Insets show the (log-)weight-versus-fiber length relationships (blue point clouds) and the interhemispheric cross correlations for the proportion of randomized edges denoted by the arrows. Lines are an average over an ensemble of 10 random surrogates at each point.

These results suggest that it could be the local background spatial 'footprint' of the empirical connectome that enables wave dynamics. To test this more directly, we performed additional simulations for a pure (deterministic) exponential weight-distance relationship, and varied the exponential decay rate. These synthetic networks again produce waves and metastable dynamics, qualitatively similar to the real connectome. Quantitatively, these dynamics are more synchronized than in the real network, implying that there are important properties of the real connectomes that are not captured by simpler surrogate networks. We also verified that wave dynamics emerge for geometric surrogate networks preserving both the original weights and weight-distance relationship but are otherwise randomly wired (Roberts et al. 2016). We have added a new Figure 10c and accompanying text on p23:

To further understand the principles of wave generation, we next generated synthetic networks with a purely-exponential weight-distance relationship (Fig. 10c). We find that this pure exponential network generates metastable waves, with slightly higher synchrony than the real brain ($R_{local}=0.95$, $R_{global}=0.61$), and similar lifetimes (Fig. 10c, left inset). Indeed waves are also found for geometric surrogates that preserve the weight-distance relationship but are otherwise random [53], exhibiting waves with $R_{local}=0.92$ and $R_{global}=0.60$. To see how the waves depend on this exponential relationship, we tested a range of slopes (i.e., characteristic lengths) between the empirical best fit and the all-to-all network with no spatial embedding. Note that again this has the effect of progressively increasing the strength of long-range connections. Similar to the partial randomization case, we find that increasing the characteristic length of the network connectivity increases the likelihood of long dwell times and eventually causes the network to synchronize. These results suggest that the key feature of the human connectome for generating waves is its spatial embedding, such that having short-range connections stronger than long-range ones facilitates the spatially-localized synchrony required for waves. However the specific properties of the waves on empirical networks – their spatial distributions of sources and sinks etc. – depend upon the specific configuration of the empirical connectome and cortical geometry.

Figure 10c: Waves in synthetic networks with pure exponential weight-distance relationship. Lines are synchronization order parameters R_{local} (solid) and R_{global} (dashed) as a function of the slope of the linear $\log_{10}(\text{weight})$ -vs-fiber length relationship. Insets show the (log-)weight-versus-fiber length relationships for the synthetic exponential networks (blue) and the original network (gray), and the interhemispheric cross correlations for the slope values denoted by the arrows.

Does the structure determine the patterns that emerge?

To an extent yes, the structure determines the patterns because random structures do not exhibit the same patterns, as per the new analyses above. Moreover, we find a relationship between hub status and source-sink densities (Fig. 7e and f). Further work would be required to comprehensively address all aspects of how the specific sequence of observed patterns depends on the network structure (e.g., pattern classification and tunable network models).

Why are the metastable states so brief (100 ms)? What precipitates the transitions?

The preceding analyses show how increasing the synchronizability of the connectome structure – such as adding long-range connections – increases the dwell time durations. There will be some other straightforward dependences on the neural mass membrane time scales and the inter-node delays. From a dynamical perspective, the transitions result from the tension between different spatial regions attempting to synchronize their phases with their locally connected neighbors, with each transition (and hence lifetime) precipitating from this tension giving way to a new dynamically-evolving phase synchrony pattern. We have added the following text to p28:

The metastable lifetimes depend on the synchronizability of the network (Fig. 10) as well as the neural mass membrane time scales and inter-node delays. The transitions result from the tension between different spatial regions attempting to synchronize their phases with their locally connected neighbors, with each transition (and hence lifetime) precipitating from this tension giving way to a new dynamically-evolving phase synchrony pattern. The situation is analogous to chimera states and metastability in other nonlinear dynamical systems coupled in a spatially dependent manner [43, 75].

There are 75 subjects with tractography, are the dynamical patterns the same for all of them, or do differences matter?

We have now simulated waves in all 75 subjects, using the subject-specific connectomes for the connectivity. This new analysis showed that all 75 subjects exhibit waves for the same $c=0.6$, $\tau = 1$ ms parameters as the group average connectome. Moreover, all 75 subjects exhibited similar distributions of dwell times and similar distributions of velocities. There is some variability across subjects, suggesting that individual connectomes do make a difference. These results are presented on p21:

Notably, waves arise on the connectomes from individual subjects, with similar speeds (subject mean \pm SD = 33 ± 2 m s⁻¹) and similar dwell times (subject mean \pm SD = 92 ± 8 ms) across the cohort (Supplementary Figure 5a,b). These distributions are fairly narrow, but there is some variability due to the individual-subject connectomes. Future work will be needed to determine how the details of the patterns vary across subjects as well as the impact on wave properties of connectomic disturbances in brain disorders.

Supplementary Figure 5a,b: Single subject wave statistics for $c=0.6$, $\tau = 1$ ms. (a) Histogram across subjects of mean speeds across all regions and times. (b) Histogram across subjects of mean dwell times.

Once a pattern transitions, does it later reappear?

This is an intriguing point. Indeed, we found that wave patterns often reappear. In fact, these recurrent patterns can be interpreted as wave motifs (Mohajerani et al., Nature Neurosci 2013). We have added the following analyses (new Fig. 5) and text to the revised MS (p9):

Visual inspection of the wave dynamics shows that the same classes of waves (traveling, rotating, etc.) frequently reappear, but the precise directions of propagation and spatial configurations appear to vary. Do specific patterns recur [4, 33]? To test this, we calculated the alignment of each node's velocity field with all of its past and future states. The average of this alignment over nodes gives an estimate of the recurrence of a specific wave pattern between different time points. We hence find that wave patterns do frequently reappear (Figure 5a). Moreover, they appear to do so more strongly than for linear surrogate time series (Figure 5b). To statistically test for such recurrences, we generated recurrence plots from an ensemble of linear surrogate time series (that preserve the amplitude distribution and linear spectra and cross-spectra but destroy nonlinear structure) and compared the ensuing distribution of whole-brain recurrence values with the empirical time series (Figure 5c). Doing so confirms the substantially larger (correlated and anti-correlated) recurrences of spatiotemporal wave patterns and, through statistical thresholding, permits formal identification of when these occur (Figure 5d).

Figure 5: Recurring flow patterns. (a) Recurrences between the mean alignment of velocity fields \mathbf{v}_i and \mathbf{v}_j at times t_i and t_j , respectively, in the model wave dynamics for $c=0.6$, $\tau = 1$ ms. (b) Recurrences for one instance of an amplitude-adjusted Fourier surrogate time series derived from the simulation used in panel a. (c) Histograms of recurrence values for the model (blue) and one surrogate (red). (d) Recurrence points where the model recurrence alignment was greater than (red) or less than (blue) all 100 surrogates. Red points correspond to flows aligned in the same direction, while all blue points correspond to flows aligned in opposite directions.

2. Maybe I missed this, but I did not understand how the results from the different subjects (75) are averaged. Were dynamics simulated on each patient's connectome? How were the results averaged? Were the simulations run multiple times with different initial conditions, and the results averaged across runs?

The simulations in the original submitted manuscript all used a group-average connectome, with the individual connectomes playing no further role. We now clarify this on p29:

Results refer to a group-average connectome unless otherwise indicated.

The dynamics were not averaged across connectomes, and most results shown were for single long runs rather than ensembles of runs with different initial. In this revised manuscript, we have added single-subject simulations (as elaborated upon in the response above, and presented in the new Supplementary Fig. 5a,b and text on p21). We have also added new analysis showing how the dwell time distributions and velocities for the strong-coupling smooth-waves case are consistent over an ensemble of 100 random initial conditions (new Supplementary Fig. 5c,d and text on p21):

We also verified robustness to initial conditions. Using an ensemble of 100 random initial conditions and the group average connectome with $c=0.6$, $\tau = 1$ ms, we observed waves with little variability across the ensemble in the nodal mean speeds (ensemble mean \pm SD = 35.3 ± 0.3 m s⁻¹) and dwell times (ensemble mean \pm SD = 89 ± 2 ms) (Supplementary Figure 5c,d). Thus the choice of initial conditions only makes a small contribution to the summary statistics of these waves.

Supplementary Fig. 5c,d: Wave statistics across an ensemble of 100 random initial conditions for the group average connectome and $c=0.6$, $\tau = 1$ ms. (c) Histogram of mean speeds across all regions and times. (d) Histogram of mean dwell times.

3. The simulated metastable dynamics are fast. The simulations are compared with slow fMRI results. Despite the “empirical evidence that short-lived states visible in electrophysiology do leave an imprint on the correlations between regions at time scales visible to fMRI”, the model results may be better compared with the MEG data. Could other features like wavelength, speed, and FC be compared between the simulation and the MEG data, in addition to the dwell time reported in Figure 2? The sources and sinks in the model are relatively stable on time scales of ~ 100 ms, and does this also occur in MEG data? This would also avoid the need to convolve the simulated data with a hemodynamic response function.

We have added a new analysis comparing the model FC to the MEG data FC. We find that amplitude envelope correlations in the MEG data correlate with those in the model ($r=0.34$, $p<10^{-15}$). This is similar to the degree of agreement found with the much slower fMRI data. We present this new result on p12:

In addition, we compared the model FC to FC derived from MEG data. We calculated alpha band (8-13 Hz) amplitude-envelope correlations from our model time series, following a recent study [51]. Again we found that the model FC correlated with the empirical FC ($r=0.34$, 95% CI [0.31, 0.38],

two-sided $p < 10^{-15}$). Thus through the faster lens of MEG amplitude correlations, our metastable waves again recapitulate the FC observed empirically.

And present the accompanying methods in new text on p32:

To compare the model FC with FC derived from MEG data, we used the publicly available amplitude-envelope correlations in Ref. [51]. We used the same approach as Ref. [51] to derive the amplitude-envelope correlations from the model: Briefly, we mapped our 513-node parcellation onto the 68 node parcellation used in the MEG analysis by taking a weighted average within each coarse parcel (weighted by overlap). We then bandpass filtered to 8-13 Hz, orthogonalized the signals, calculated the Hilbert amplitude envelopes, downsampled to 1 Hz, and calculated Pearson correlations between all pairs of amplitude time series. We then compared our model FC to the group-average FC (across 55 subjects) in the MEG dataset.

Applying the wave analyses (velocity, streamlines, etc.) to the MEG data would be a major undertaking because applying appropriate constraints on source reconstruction (such as the nature of the expected zero and non-zero lag covariances) that respect wave patterns will require substantial new theoretical work that we aim to achieve in a follow-up study. We have added this observation to an existing commentary on such challenges (p26),

For human neuroimaging, progress will require source reconstruction techniques that can appropriately accommodate the complex amplitude-phase relationships and lagged covariances that reflect metastable waves.

4. I was at times confused by the “functional network” terminology. From the Abstract, one claim is that this work moves “the study of functional networks from a static to an inherently dynamic frame”. I use the definition from Sporns to define a functional network as nodes with coupled activity, which requires a “dynamic frame” to observe. On page 13, the “classic functional networks” are defined differently.

Here we were referring to the recent controversy over whether functional connectivity is “dynamic” – the reviewer is of course correct that observing any functional connectivity requires dynamics in the first place. Our point is that at the moment the dominant paradigm in the functional connectivity community is to summarize the FC by a single time-averaged matrix, with growing interest in whether this subsumes more interesting dynamics of the correlation structure itself across time. Moreover, even the work on dynamic functional connectivity that permits briefly expressed “meta-states” assumes that activity within each state is spatially stationary – such as stationary cluster states. Our work argues that these meta-states may include “inherently dynamic frames” – i.e., waves.

We have now clarified the definition of “classic functional networks” on p15:

*Do these preferential sites of sources and sinks overlap **with canonical functional subnetworks (that possess strong internal functional connectivity)?***

Moreover, we have reworded the “functional network” in the abstract to state:

*By moving the study of functional networks from a **spatially static** to an **inherently dynamic (wave-like)** frame, our work unifies apparently diverse phenomena across functional neuroimaging modalities and makes specific predictions for further experimentation.*

Minor comments

“Yet with the exception of the pathological strongly-nonlinear dynamics in epileptic seizures [5, 6], theory and modeling have fallen behind the body of empirical results.” For more specific references to waves in seizures see (Smith, E., Liou, J.-Y., Davis, T., Merricks, E., Kellis, S., Weiss, S., et al. (2016). The ictal wavefront is the spatiotemporal source of discharges during spontaneous human seizures. *Nature Communications*.) and (Martinet, L.-E., Fiddymment, G., Madsen, J. R., Eskandar, E. N., Truccolo, W., Eden, U. T., et al. (2017). Human seizures couple across spatial scales through travelling wave dynamics. *Nature Communications*, 8, 14896.)

We now cite these papers on p2:

*Waves ... have also been implicated in sensorimotor processing of saccades [29], **propagating seizure fronts [30, 31]**, and observed during sleep with a possible role in memory consolidation [4].*

pg 4, Why is the nodal speed slower than the wave propagation speed?

The wave propagation speed histogram is across all regions and times, whereas the nodal speeds are averages over time and so will be lower than the highest speeds observed at each region. We have clarified this on p4:

*Average nodal mean speeds range from 23-52 m s⁻¹ (Fig. 2b), **slower than the nodal maximum speeds and considerably narrower than the full set of instantaneous speeds.***

Figure 2, Are these results for one simulation? Or for many simulations with different initial conditions?

These results are for one simulation, but as detailed above we have added an analysis for different initial conditions in the new Supplementary Fig. 5c,d.

Figure 2, Can nodal speed be related to structure connections? Would a plot of nodal speed vs structure network degree indicate this? My naive guess is that speed is related to structure in some way, since that's the only feature that differs between the nodes.

We have added a new plot of nodal speed vs node degree (Supplementary Fig. 1) and added the following to the text on p5:

Nodal speed is associated with node degree such that hubs tend to support lower speeds than peripheral nodes ($r = -0.17$, $p = 8.1 \times 10^{-5}$; Supplementary Figure 1), consistent with heterogeneity of time scales found in prior work [48].

Supplementary Figure 1: Correlation between nodal mean speed and node degree. Degrees are calculated for the network thresholded to 30% density. Line is a least-squares fit.

Note that nodes also differ in the geometric location and this also impacts upon speed (Fig. 2C).

pg 7, “calculate the instantaneous coherence”, I think the measure used is the phase locking value (PLV).

As we understand it, phase-locking value is usually defined between two time series, whereas this instantaneous coherence is for a single time series (namely, the mean of all phasor time series within a hemisphere). However, we had written the definition in Eq. (7) as a sum rather than an average, which we have now corrected.

pg 7, “At the time of a metastable transition, the large-scale wave pattern breaks up and disorganized short wavelengths dominate. Thus, the metastable wave signatures are separated by narrow periods of time with relatively low correlation between the hemispheres.” This conclusion is not clear in Figure 3. How do we see short wavelength and low correlation? How were black lines chosen in Figure 3b?

In Figure 3a (lower panel), low correlation is given by the points in time (vertical slices in the plot) with $cc \sim 0$ for all lags (i.e., the vertical slice is predominantly green), indicating that those time points lack a strong correlation structure (strong correlation corresponds to red and blue banding). We now clarify this on p8:

*We use these brief periods of phase desynchronization, yielding low values of the interhemispheric cross-correlation function, to identify transitions: **This is achieved by thresholding the cross-correlation function when it is close to zero for all time lags** (vertical black lines in Fig. 3b; see **Methods**).*

And the Methods on p33:

Time series of the time-lagged cross correlation were thresholded to identify transitions between different spatiotemporal patterns. Transitions correspond to low cross-correlation for all time lags. We calculated $1/\text{Var}[C(t,l)]$, the inverse variance across lags l at each t . We then thresholded this quantity by calculating its mean and finding all suprathreshold time intervals. Transition times

correspond to the peak in $1/\text{Var}[C(t,l)]$ within each suprathreshold interval. The dwell time distributions are robust to modest changes in the value of this threshold.

pg 8, “Extracting all times corresponding to low synchronization thus yields an automatically-computed set of transition times”, please define “low”, my apologies if I missed this.

As per the previous response, we have clarified this on p8 and p33. Note that the ensuing results are robust to the exact choice of this value.

pg 10, “Our model thus provides a plausible mechanism for these metastable transitions.” More accurate to state that the model captures one aspect of the metastable transitions, the dwell time.

We have edited the sentence (on p11) to now read:

*Our model thus provides a plausible mechanism for **the dwell times** of these metastable transitions.*

pg 10, Is the r computed for one subject, or for all subjects? Is FC computed for the entire trace, or for different stable intervals of the dynamics? Is the FC similar because both model and rsfMRI have same SC?

The r was computed between the model FC using the group-average-connectome and the group-average empirical FC. The FC is computed for the entire trace. It would indeed be expected that some component of the FC similarity will be driven by the common SC, as shown in recent studies into the relationship between SC and FC (e.g., Honey et al., Messe et al., Robinson). We now clarify that the empirical FC are group averages of long-time FC (p11):

*We thus asked what our metastable waves would look like through the standard lens of **static** resting-state functional connectivity (FC) as studied in fMRI.*

and

*...**long-time** resting-state fMRI **averaged over** same group of subjects...*

and mention that SC plays a role (p12):

Note also that the shared structural connectivity between the model and the fMRI data would likely contribute to this correlation [49, 61, 63].

Figure 5 caption, “colored by orientation in the plane)”, is the orientation in 3 dimensions, not 2 dimensions?

This is the orientation in 2-D as viewed from the top. We have edited the caption (now Fig. 6) to state: “*colored by orientation in the 2-D plane shown*”.

pg 12, “Even though we have not imposed any temporal smoothness (e.g. as done in optical flow methods [15]), the streamlines are well- behaved across time, reflecting the stability and order of the underlying activity patterns.” The instantaneous phase at each node is extracted using the Hilbert transform, which may impose temporal smoothness.

We have clarified that we mean that we have not imposed temporal smoothness in the velocity or streamline estimation using a regularization parameter or low-pass filter or similar. The Hilbert transform does not impose temporal smoothness per se, although to permit a well-defined instantaneous phase on noisy stochastic time series, the use of a prior bandpass filter is widespread in the field. Added to p14:

Even though we have not imposed any temporal smoothness in the velocity or streamline estimation (e.g. using filters or regularization as done in optical flow methods [15])...

pg 12, “The first (Fig. 5b, left) shows two frontal left-hemisphere sources and a sink that merge and reconfigure to yield one left frontal source, while in the right hemisphere one frontal source and one frontal sink move closer together and a temporal sink emerges. The second example (Fig. 5b, right) exhibits complex frontal clusters of sources and sinks that split and move posteriorly, plus the emergence of an occipital sink.” Difficult to see these claims in Figure 5.

We have added arrows to the figure to highlight these features we refer to:

*The first (Fig. 6b, left) shows two frontal left-hemisphere sources and a sink that merge and reconfigure to yield one left frontal source (**green arrows**), while in the right hemisphere one frontal source and one frontal sink move closer together (**purple arrows**) and a temporal sink emerges (**yellow arrow**). The second example (Fig. 6b, right) exhibits complex frontal clusters of sources and sinks that split and move posteriorly (**dark red arrows**), plus the emergence of an occipital sink (**light blue arrow**).*

pg 13, “calculating the standard deviation of visits to each node”, Visits of what to each node? What is the standard deviation of visits, is it a standard deviation of counts? In 20 ms, seems like very short window to capture variability of spatial dynamics?

By a visit to a node we mean that a source (or sink) is, at one time point, nearest to that node. Across time, this number varies. The standard deviation of visits for a given node is hence the standard deviation across time of the number of visits to that node. The 20 ms window is used specifically to detect rapid reconfigurations, but indeed one could measure the variability of spatial dynamics by calculating the same quantity over a longer duration. We have clarified the text on p14 to now read:

To quantify this, we estimated the moment-to-moment temporal variability of sources (and sinks) by calculating the standard deviation of the number of visits of sources (and sinks) to each node, and averaged this across nodes. We used short sliding windows (non-overlapping of length 20 ms), to specifically detect rapid reconfigurations.

pg 14, “while sources visit the somatomotor hand network more often than others including the default mode.”, That’s not completely clear from the figure, the distributions appear to overlap quite a bit. Maybe the mean value of fraction of visits is different.

Indeed the mean value of fraction of visits is different, we now clarify on p15 that there is overlap between the distributions:

*For example, **while there is substantial overlap**, the ventral attention subnetwork is visited by sinks **on average** more often than each of cingulo-operculum and subcortical, while sources visit the somatomotor hand network more often than others including the default mode.*

pg 14, “This analysis confirmed that the feeder nodes act as sinks significantly more often than hubs and non-hubs (Fig. 6e), while the hubs act as sources significantly more often than the non-hubs (Fig. 6f).” p values? Also the effect could be significant but the difference appears to be very small.

The p values are given in Supplementary Table 2, which we now reference in the text on p16. And yes as the data points show, the effects are not large; we now note this on p16:

***While there is substantial overlap**, this analysis confirmed that the feeder nodes act as sinks significantly more often than hubs and non-hubs (for details, see **Supplementary Table 2**; Fig. 7e), while the hubs act as sources significantly more often than the non-hubs (Fig. 7f).*

pg 18, The first paragraph suggests the results are robust to changes in network structure. But it’s not quantitative. The movie does provide evidence that waves exist in different network structures. But it’s not clear that the waves are “similar to those in our fully-connected weighted connectome”. That’d require more quantitative analysis. The second paragraph suggests the results are consistent to changes in the neural mass model. This is an interesting and important observation, but the discussion is suggestive and the results not quantitative.

As outlined in the response to Point 1 above, we have added extensive new quantitative analyses for other network structures with increased randomness in the new material on pp21-23. For the thresholded networks in Supplementary Movie 9, we now also quantify their velocities on p21:

*Waves similar to those in our fully-connected weighted connectome exist also in sparser networks (thresholded down to 10% density), and occur whether using traditional weight-based thresholding [67] or consistency-based thresholding [68] (**mean nodal mean speeds 27 m s⁻¹ and 29 m s⁻¹, respectively**). Waves exist also on a 10%-density binary network thresholded by weight (with all weights set to 1; **mean nodal mean speed 33 m s⁻¹**).*

Further exploration of other models is beyond the scope of this work, but we now mention that our comparison is qualitative and that future work will be needed to quantify the diversity of wave dynamics there (p27):

Likewise, we also showed that waves are not specific to the neural mass model we employed but also arise on the human connectome when using the Wilson-Cowan model [62] and the Kuramoto model [64, 65]. These models fundamentally differ in the nature and time scales of their internal dynamics, the degree of mathematical and physiological abstraction, and the mechanism by which coupling is

facilitated. Prior studies of the Kuramoto model have shown that time delays and spatially-constrained connectivity can engender the sort of multi-frequency effects presently observed [64, 65, 75]. In addition, although not formally analyzed a recent study of brain eigenmodes using the Wilson-Cowan model also appears to support the emergence of traveling waves [39]. The presence of waves across all three models speaks to principles that are not dependent upon particular choices of models or their parameters, although future quantitative analyses are required in order to understand how these particularities influence the time scales and other properties of the ensuing waves.

pg 19, “While metastability and waves in neural systems have been studied in isolation, here we show for the first time that these dynamical regimes are compatible, can arise from the human connectome, and can be explained with a unified mechanism.” Please review the unifying mechanism in the Discussion.

We have added the following to the Discussion (p25):

Moreover, we observe that metastable transitions in our simulations arise through the collision, annihilation and/or creation of new sources and sinks. These dynamics underlie the co-occurrence of waves (the nature and distribution of these nodes) and metastability (their collision), hence providing a unifying mechanism for their co-occurrence.

pg 19, “This suggests that waves have functional specificity.” This is not clear from Figure 6c,d.

We have reworded the Discussion to now read:

Although there is substantial overlap, this suggests that waves may have functional specificity in the resting state. For instance, one subnetwork in which sinks tend, on average, to congregate is the default mode, consistent with its purported unique role in task-free/resting state acquisitions. Task constraints may lead to a reorganization of wave patterns with a consequential increase in the location and functional specificity of sources and sinks.

pg 19, “one subnetwork in which sinks tend to congregate is the default mode, consistent with its purported unique role in task-free/resting state acquisitions.” this is very difficult to see in Figure 6c, the DM result does not stand out.

We have reworded this point, as above.

pg 23, Equation 1, Please verify inhibitory input is a function of inhibitory activity and inhibitory membrane potential?

This is correct, as now noted below the equations:

Note that in Eq. (1), inhibitory input is a function of inhibitory activity and inhibitory membrane potential.

pg 24, “are the corresponding thresholds for axon potential”, action potential.

Fixed.

pg 26, “We define the interhemispheric synchrony as the sliding-window time-lagged cross-correlation (t ,) between (t) and (t).” This measure is the cross-correlation of the phase locking values PLV. For “interhemispheric synchrony”, why not compute PLV between brain regions across hemispheres?

As per the response above, the phase coherence of a group of signals is not the same as the PLV between a pair of signals, though the expressions are similar. And indeed one could calculate PLV between all the interhemispheric pairs and then calculate an average interhemispheric PLV, but we have not done so here.

Reviewer #2

In this beautiful paper, the authors investigate a whole-brain model that exhibits propagating wave patterns similar to experimental findings. The authors show, by simulating resting state conditions, that different wave patterns are seen in succession, and they form metastable states (different from attractor states). The paper also proposes ways to reconcile different observations, and opens many avenues for further research, so it is a very useful contribution to the field.

We thank the reviewer for this positive appraisal of our work and constructive feedback. Our detailed responses are given below, interleaved with the reviewer's comments (in blue).

I found that three aspects should be improved:

1. The authors should relate their work to that of Tim Murphy's lab (Mohajerani et al., Nature Neurosci 2013) where they found that motifs of activity (sometimes propagating) relate to axonal projections in mouse. They also found that propagating waves are all over the brain and many times symmetric between hemispheres (Mohajerani et al. J Neurosci 2010). Since some of these findings are similar to what is reported here, it would make sense to discuss the similarities and differences.

We now discuss these papers on p25:

Wave sources and sinks also overlapped heterogeneously with functional network properties. Although there is substantial overlap, this suggests that waves may have functional specificity in the resting state. For instance, one subnetwork in which sinks tend to congregate is the default mode, consistent with its purported unique role in task-free/resting state acquisitions. Task constraints may lead to a reorganization of wave patterns with a consequential increase in the location and functional specificity of sources and sinks. These observations are consistent with the presence of waves of rodent cortex, as evident in VSD recordings, exhibiting heterogeneous spatial distributions including patterns of sources and sinks [33] and bilateral symmetries [34]. These patterns reorganize upon brief whisker stimulation, and include recurring sensory motifs embedded in spontaneous activity [33]. Moreover, wave patterns overlap with patterns of long-range structural connectivity, suggesting a role for the connectome in shaping the dynamics.

We also cite these papers at other points in the manuscript as relevant.

2. As the authors acknowledge, there is no noise considered, which seems to be justified by the fact that the "resting state" is supposed not to be influenced by external inputs. However, this would be true for deep slow-wave sleep, but I do not think one can consider the resting state as totally isolated. The authors should include a simulation, even very preliminary, in the presence of weak noise, to show how the propagating patterns are affected. This is particularly important for transient (metastable) states, which could be very sensitive to noise. Is it the case here ?

Following this suggestion, we have undertaken new analyses that show that the existence of waves is robust to the inclusion of noise, as is the existence of metastable states. The corresponding wave patterns are less locally coherent than in the noise-free case. Interestingly, weak noise has a stabilizing effect and increases global synchrony and dwell times. Stronger noise breaks the local synchrony and

eventually extinguishes the waves at fixed coupling strength. This new analysis is shown in the new Supplementary Figure 6 and Supplementary Movie 10, described in the text on p24:

Additionally, we verified that wave patterns persist in the presence of weak noise (Supplementary Movie 10). These wave patterns have lower local synchrony than the noise-free case but also exhibit metastability (Supplementary Figure 6). Interestingly, weak noise also increases the global synchrony and dwell times, similar to the effects of stochastic resonance in other systems [71]. Further increases in noise decrease local synchrony and extinguish the waves (at fixed coupling).

Supplementary Figure 6: Metastable transitions exist in the presence of noise. **(a)** Local and global synchrony as a function of additive noise amplitude σ . Lines show synchronization order parameters R_{local} (solid) and R_{global} (dashed). **(b)** Interhemispheric cross-correlation functions for the six values of σ indicated by arrows. **(c)** Mean dwell time as a function of σ . Note the slight decrease in local synchrony, increase in global synchrony and longer dwell times for weak noise, $\sigma \leq 0.002$. Simulations performed for $\tau = 0, c = 0.6$.

Simulation details are given in the Methods on p31:

Noise simulations were performed by replacing the input current term $a_{ne}I_0$ in Eq. (1) with $a_{ne}[I_0 + \sigma\eta(t)]$, where $\sigma\eta(t)$ is zero-mean Gaussian white noise with standard deviation σ [83]. The model was solved for $\tau = 0$ using the Heun scheme with a time step of 0.01 ms.

3. The findings are here reported using only one particular model, which is a neural mass model introduced by the authors. To what respect the findings depend on this model, or would be general, do you have any evidence for this? (like a previous paper where the model was compared to another one?)

We did indeed show that waves exist in other models, but have added further elaboration of this, also in response to R1 (p28):

Likewise, we also showed that waves are not specific to the neural mass model we employed but also arise on the human connectome when using the Wilson-Cowan model [62] and the Kuramoto model [64, 65]. These models fundamentally differ in the nature and time scales of their internal dynamics, the degree of mathematical and physiological abstraction, and the mechanism by which coupling is facilitated. Prior studies of the Kuramoto model have shown that time delays and spatially-constrained connectivity can engender the sort of multi-frequency effects presently observed [64, 65, 75]. In addition, although not formally analyzed a recent study of brain eigenmodes using the Wilson-Cowan model also appears to support the emergence of travelling waves [39]. The presence of waves across all three models speaks to principles that are not dependent upon particular choices of models or their parameters, although future quantitative analyses are required in order to understand how these particularities influence the time scales and other properties of the ensuing waves.

Minor:

On page 20, when you say that high time resolution is used, citing only MEG and ECOG, you should also include VSD, because many traveling wave studies have been using this technique.

We now mention VSD on p26:

...such as MEG, ECoG, and VSD.

See also our response to your first point.

Reviewer #3

In this work the authors show the emergence of various kinds of wave dynamics in the brain at macro-scale using a neural mass model applied to the structural connectivity of the brain. Quite remarkably, the model dynamics not only show various types of waves, e.g. travelling, rotating, spirals, sources and sinks but also exhibit spontaneous transitions between these waves, even in the absence of noise. Such transitions provide further evidence for metastable dynamics, winnerless competition between different states or attractors, which is now increasingly often hypothesized and shown to underlie the brain dynamics. The metastable transitions observed in the numerical simulations of the model utilized here occur via brief desynchronisation periods during which wave patterns reconfigure, which the authors automatically characterize via interhemispheric cross-correlation.

Overall, I quite enjoyed reading this manuscript and I think the demonstration of the emergence of and metastable transitions between various wave dynamics, which have already been observed in empirical MEG data, is an important contribution. Below are some comments and questions that can improve the clarity of presentation of this manuscript and help better relate it to the existing models in the literature.

We thank the reviewer for this positive appraisal of our work and constructive feedback. Our detailed responses are given below, interleaved with the reviewer's comments (in blue).

Major comments:

1. I think my main major comment is about tuning the model parameters to fit the empirical data. As one of their key contributions, the authors state on page 4 “We explore a range of coupling strengths and delays beyond the narrow area of parameter space previously studied”. My main question is whether that parameter space is biologically realistic. Generally most computational models are applied by optimizing the model parameters to best fit the observed functional connectivity or some other characteristic of the fMRI or MEG data (e.g. Deco & Kringelbach 2014 Neuron, Cabral 2011 Neuroimage, Cabral 2014 Neuroimage, Deco 2017 Neuroimage). I find the wide range evaluation of the parameter space that the authors present, quite important in terms of understanding different dynamics the model is capable of creating, but I think it is also important to find out, which of the simulated dynamics are biologically meaningful, e.g. they exhibit similar characteristics as the empirical data in fMRI, MEG or functional connectivity matrices. For instance on page 17 Figure 7a, some of the presented functional connectivity matrices seem to be quite different than that of the empirical data. Can the authors please clarify this?

This is a good point. We have added a new Figure 9 showing the correlation between the model and empirical functional connectivity values (with and without GSR). Comparing this figure with Fig. 8a, we see that the dynamical regimes that best agree with the empirical values are those that have a hemispheric block structure in their FC. We also note that the best fit was found for $c=0.2$, $\tau = 0$ ms, though the difference is small vs several other regions, and in particular the smooth waves $c=0.6$, $\tau = 1$ ms case has the second highest correlation across the ranges tested, both with and without GSR. We discuss the new figure in text added to p20:

Prior studies modeling neuronal dynamics on empirical connectomes have used the match between predicted and empirical functional connectivity to tune underlying parameters accordingly [62, 64-66]. We hence tested how spatial waves compared to these other candidate spatiotemporal patterns in

their match to empirical resting state functional connectivity (Fig. 9). Intriguingly we find that the smooth waves observed for the parameter combination of $c=0.6$ and $\tau=1$ ms yield the second highest correlation across the entire parameter space tested (with or without GSR), only marginally lower than the best global fit (for $c=0.6$ and $\tau=1$ ms). Additional tuning of the model parameters could improve this fit. Moreover, triangulating model fit with other dynamic metrics – wave properties, dwell times, source/sink distributions – would improve the identifiability of the model parameters from empirical data.

Figure 9: Correlation between modeled and empirical functional connectivity as a function of coupling and delay. (a) Pearson correlation between empirical and modeled FC values for each pair of regions, without GSR. Numbers 1-3 indicate the top three highest correlations. (b) Same as (a) but with GSR.

2. The paper generally presents a thorough literature study, but I think the following references are also very relevant for this study: (Cabral 2014 Neuroimage) and (Deco 2017 Neuroimage), where the authors model MEG data using a Kuramoto and a multi-frequency Hopf model, respectively. Also, in reference 35, the authors also utilize a Wilson-Cowan model, and their videos show rotating as well as travelling waves. Although it has not been explicitly discussed as travelling wave dynamics, I think their supplementary videos and numerical simulations of Wilson-Cowan equations support the findings of this manuscript quite nicely. The authors may want to refer to these findings using the Wilson-Cowan model, in relation to their findings with this model, e.g. on page 18 last paragraph before Discussion.

We have followed this suggestion and now cite and discuss Cabral et al. 2014, Deco et al. 2017, and the videos of Atasoy et al. 2016 on p28:

Likewise, we also showed that waves are not specific to the neural mass model we employed but also arise on the human connectome when using the Wilson-Cowan model [62] and the Kuramoto model [64, 65]. These models fundamentally differ in the nature and time scales of their internal dynamics, the degree of mathematical and physiological abstraction, and the mechanism by which coupling is facilitated. Prior studies of the Kuramoto model have shown that time delays and spatially-constrained connectivity can engender the sort of multi-frequency effects presently observed [64, 65, 74]. In addition, although not formally analyzed a recent study of brain eigenmodes using the Wilson-Cowan model also appears to support the emergence of travelling waves [39]. The presence of waves across all three models speaks to principles that are not dependent upon particular choices of models or their parameters, although future quantitative analyses are required in order to understand how these particularities influence the time scales and other properties of the ensuing waves.

3. The states in the MEG data are characterized using a hidden markov model (HMM) with 12 states (page 9). How is the number of states chosen, how would it affect the distribution to change the number of states in the HMM?

The number of states was inherited from Vidaurre et al. 2018 where the HMM states were estimated. We now clarify this on p32:

The transition times for the HMM were as derived in [60]. In general, the number of states determines the level of detail in the analysis, such that increasing the number of states causes states to split, yielding a hierarchical view of the data. Twelve was chosen without any claim that this is the biological truth, but with checks for reliability of the state assignments across half-splits in the data. Alternative approaches include use of free energy for model selection purposes in a manner that balances model likelihood with model complexity.

Minor Comments:

4. Page 5: Can the authors comment or speculate on what may be the reason that the travelling wave speeds increase with distance from midline (Page 5 and Fig. 2)?

There are likely numerous geometric and topological reasons. One contributor would be the corresponding preference of sources in similar midline regions. We now note that (p13);

... This preference for midline sources may partly underlie the distribution of nodal speeds, which are typically low in the same regions (Fig. 2d) because wave fronts propagate slowly in the vicinity of a source and gather speed further away.

5. Page 5: “The regions supporting faster wave fronts from a roughly contiguous zone in each hemisphere, spanning from the frontal lobe to the occipital lobe, via the parietal lobe and posterior aspects of the temporal lobe.” How to these regions relate to sensory or higher level cortices?

We now report the functional network membership of the top ten fastest and slowest nodes on p6:

To place this in terms of the classic functional networks, we assigned brain regions to twelve subnetworks (somatomotor hand, somatomotor mouth, cingulo-opercular, auditory, default mode, memory, visual, fronto-parietal, salience, subcortical, ventral attention, and dorsal attention) according to a broadly-used functional subdivision of the brain [58] (Supplementary Figure 2). The top ten fastest nodes lie in the somatomotor hand, auditory, default mode, fronto-parietal, and ventral attention networks. The top ten slowest nodes lie also in the default mode (thus indicating a wide diversity in its wave speeds), plus memory and visual regions.

6. How are the low synchronization time points identified; e.g. using a certain threshold, if so what is the threshold? For instance in Fig. 3 caption, the authors state: “(b) Vertical lines depict instances of low values of the interhemispheric cross-correlation function, corresponding to wave transitions”. What threshold is considered to be low?

We have clarified this in the Methods on p33:

Time series of the time-lagged cross correlation were thresholded to identify transitions between different spatiotemporal patterns. Transitions correspond to low cross-correlation for all time lags. We calculated $1/\text{Var}[C(t,l)]$, the inverse variance across lags l at each t . We then thresholded this quantity by calculating its mean and finding all suprathreshold time intervals. Transition times correspond to the peak in $1/\text{Var}[C(t,l)]$ within each suprathreshold interval. The dwell time distributions are robust to modest changes in the value of this threshold.

7. Fig 5: It would be clearer if the authors state that x represents the distance to the midline in Fig 5c, in the figure caption.

We have edited the Fig. 6 caption accordingly:

Lateral positions (displacement from the midline, x) of sinks (top) and sources (bottom) plotted across time, colored by vertical (dorsoventral) position z .

8. Can the authors please comment on why the sources and the sinks have the tendency to remain within a single hemisphere (page 13, end of first paragraph)? Would that be due to the relatively smaller number of interhemispheric connections in DTI?

This is an interesting question. We now consider this on p15:

Moreover, there is a tendency for sources and sinks to remain within a single hemisphere (i.e., the trajectories traced by colored dots in Fig. 6c tend not to cross $x=0$). This may reflect the relatively weaker interhemispheric connectivity acting as a barrier to sources and sinks traversing the hemispheres, or because the relatively small “aperture” causes them to collide. It could also be that slowing toward the midline means they dissolve before they get a chance to cross.

9. Page 14: “This analysis confirmed that the feeder nodes act as sinks significantly more often than hubs and non-hubs (Fig 6), while the hubs act as sources more often than the non-hubs”. Can you please report the p-values or refer to Supplementary Table S2, if that is the corresponding p-values?

We now cite Supplementary Table 2 in the text on p16.

10. Page 16: “Waves are also primarily observed for stronger coupling with longer delays, occurring across a relatively large regions of parameter space (e.g. for $c=0.4-0.6$, $\tau=6-10\text{ms}$ ”. Again, would this range agree with biologically meaningful parameter values; e.g. if the model were to fit to best represent the functional connectivity of the empirical data, would the estimated parameters fall into that range?

The correlation between model and empirical FC is highest for these parameter values, as pointed above in our response to your first point.

These parameter combinations are biologically plausible; we now point this out on p27:

The best fits to functional connectivity and dwell times were observed for stronger coupling combined with either short delays ($\tau = 1$ ms) or longer delays occurring across relatively large regions of parameter space (e.g. for $c = 0.4-0.6$, $\tau = 6-10$ ms). These parameters fall within biologically realistic limits (with the stronger coupling and short delays emphasizing the role of nearby nodes).

11. Page 18: “In the Kuramoto case we observed waves when the model is in a regime of partial synchronization”. How does this range relate to the parameter range estimated by fitting the Kuramoto model to the resting state data; e.g. Cabral 2011 Neuroimage and Cabral 2014 Neuroimage, where the authors fit a Kuramoto model with delays to fMRI and MEG data?

The proof-of-principle use of the Kuramoto model follows our recent work aiming at time scales that match those of the BOLD response – similar to recent work from our own group (Gollo et al. 2017) and papers of Deco et al. and Cabral et al.:

Note that the time scales of this proof-of-principle use of the Kuramoto model are tuned to match those of the slow BOLD response – as with other recent studies [62, 95] – and hence differ from prior objectives to directly match the faster frequencies of MEG data [65]. As such, the parameterizations (coupling strength) are not directly comparable, though we note that these dynamics are in a regime where the Kuramoto order parameter has mean 0.78 and standard deviation 0.14 across the time series. Note that we do not include phase delays (equivalent to time delays) in these illustrative simulations. Prior work suggests that time delays may include additional dynamic instabilities [75] and intermittent frequency slowing [64].

12. Page 23: Please specify what c and C denote in Eqs 1,2 and what e and I denote in $x(=e,i,n)$.

Done (p30):

*Here, delayed inputs from other regions in the network enter through the term $cQ_j^{network} = c \sum_k C_{jk} Q_V(V_k(t - \tau)) / \sum_k C_{jk}$, where τ is the delay time, c is the global coupling strength, **and C_{jk} is the connectivity weight from region k to region j .***

and

*... and a_{xy} terms parameterize the strength synaptic coupling from population x ($= e, i, n$, **where e and i are the excitatory and inhibitory populations, respectively, and n is a non-specific input**) to population y ($= e, i$).*

13. Page 26: Methods, Functional connectivity: “We calculated functional connectivity both on mean pyramidal cell body potential, and after estimating the BOLD ”: which version of the functional connectivity was then used for further analysis. I thought only the one after the convolution with the hemodynamic response was used, but that is not clear in the statement above. Can you please clarify?

For the comparison with resting-state fMRI data, we only used the functional connectivity after estimating the BOLD (both with and without global signal regression). For the parameter sweep in Fig. 8, the caption was incorrect: we show the FC matrices calculated on the raw time series. We have

added the FC for BOLD without and with GSR in the new Supplementary Fig. 4, and corrected the caption:

Figure 8: Dynamics as a function of coupling strength c and delay τ . (a) Functional connectivity matrices calculated directly from the neuronal time series (corresponding results after convolution of the neuronal time series with a hemodynamic response function are provided in Supplementary Figure 4).

... and clarified the Methods text on p27:

We calculated functional connectivity both on the mean pyramidal cell body potential (Fig. 8a), and after estimating the BOLD signal (Supplementary Figure 4).

Supplementary Figure 4: FC matrices calculated on model BOLD time series. (a) BOLD FC without global signal regression. (b) BOLD FC with global signal regression.

14. Inter-hemispheric synchrony: I wonder what effect the separation of two hemispheres has for this analysis; i.e. could this synchrony measure also be applied with a different partitioning of the brain; i.e. front-to-back, and still capture the transitions, or is there a specific information in the inter-hemispheric synchrony?

Yes, this synchrony measure can be applied to any partitioning of the brain, and we now compare left-right to anterior-posterior and dorsal-ventral in a new Supplementary Figure 3 and text on p7:

Here we used an interhemispheric partition, but any partition can be used in principle. We additionally tested partitions along the anteroposterior and dorsoventral axes, and found that they also capture the transitions (Supplementary Figure 3). However, it turns out there is specific information in the interhemispheric partition: front-back and top-bottom partitions are more similar to each other than they are to the left-right partition, possibly reflecting the unique ‘gating role’ of the corpus callosum.

Supplementary Figure 3: Coherence cross-correlations for different partitions of the brain. (a) Left-right interhemispheric cross-correlation. (b) Anteroposterior cross-correlation. (c) Dorsoventral cross-correlation. Sliding-window cross-correlations are calculated on the same time series in all three panels. Node colors (right column) denote the partitions formed by median split in each direction.

15. Page 27, last paragraph, please state that C refers to the connectivity matrix, ideally first time you introduce C in Eq 1.

As detailed in the response to Point 12 above, we now define C shortly after Eq. (1), and have reiterated it here in this subsection:

... where C_{jk} is the connectivity weight from region k to region j.

16. The authors chose use the Kuramoto mode without delays, although that has been reported to be a crucial component for realistic simulations (Cabral 2011 and 2014 Neuroimage). As Kuramoto model is not the main focus of this paper, I think it is fine if the authors chose to proceed without including the delays to Kuramoto model, but I would suggest at least commenting on how that may affect the wave dynamics.

We have incorporated this comment into our response to your point 11 above,

Note that the time scales of this proof-of-principle use of the Kuramoto model are tuned to match those of the slow BOLD response – as with other recent studies [62, 95] – and hence differ from prior objectives to directly match the faster frequencies of MEG data [65]. As such, the parameterizations (coupling strength) are not directly comparable, though we note that these dynamics are in a regime where the Kuramoto order parameter has mean 0.78 and standard deviation 0.14 across the time series. Note that we do not include phase delays (equivalent to time delays) in these illustrative simulations. Prior work suggests that time delays may include additional dynamic instabilities [75] and intermittent frequency slowing [64].

17. Supplementary Fig. S1: It may help using a different colormap to better distinguish different resting state networks and improve the clarity of presentation.

We have tested several different colormaps, but due to the number of functional communities (13) and their spatial inter-mixing, it is difficult to better disambiguate these networks with different color options.

18. Supplementary Tables S1 and S2: I may be misunderstanding this analysis but can the authors please clarify the details of the comparisons presented in the supplementary tables, t-tests between exactly what quantities. I believe I fail to understand how to read the tables. In my understanding you test the significance of the overlap between a functional network and sinks (or sources) emerging within that network. Shouldn't this result in one vector of size (1 x nr of resting state networks) for the sinks and one vector for the sources rather than a matrix. Can the authors please clarify?

We have tested all pair-wise comparisons between the resting state networks, giving 13×12/2 possibilities. We have clarified this in the captions:

*Numbers are two-tailed p-values for **all pair-wise t-tests between the numbers of visits to each functional network**, corrected for multiple comparisons (Bonferroni).*

And

Numbers are two-tailed p-values for all pair-wise t-tests between the numbers of visits to each category, corrected for multiple comparisons (Bonferroni).

Reviewers' Comments:

Reviewer #1:

None

Reviewer #2:

Remarks to the Author:

The authors have provided the necessary clarifications with respect to the points I raised, and I think the paper is much improved. I have no further comment, and recommend acceptance.

Reviewer #3:

Remarks to the Author:

The revisions and newly added analysis in the updated manuscript address my questions and comments. I only have two minor questions about the newly added parts, which in my opinion would help improve the clarity of the presentation of the revised manuscript:

1) I think the fitting to the functional connectivity (Fig. 9) provides an important addition to the presented study and links it quite nicely to other studies on computational models in the literature. It is also quite important that the parameter combination yielding the smooth waves also have the second highest correlation across the entire parameter space. Can the authors please comment on the type of dynamics observed for the best fit between the model and the functional connectivity matrix, were they also wave dynamics?

Also if I understand correctly there is a typo in the newly added explanation about that: "Intriguingly we find that the smooth waves observed for the parameter combination of $c=0.6$ and $\tau=1$ ms yield the second highest correlation across the entire parameter space tested (with or without GSR), only marginally lower than the best global fit (for $c=0.6$ and $\tau=1$ ms)." I believe the values in the parenthesis should be the best global fit; i.e. $c=0.2$, $\tau=0$ ms.

2) The added analysis on the metastable transitions is also very intriguing. I think the additional finding showing that there is specific information in the inter-hemispheric connections is quite interesting. In my opinion, for the use of the partitioning in this study, the important point would be that the detected metastable transitions (Fig. 3 b) are consistent across different partitions. Could the authors please comment on whether that is the case?

Typo:

- Please define GSR in the manuscript.

Reviewer Responses

Reviewer #2

The authors have provided the necessary clarifications with respect to the points I raised, and I think the paper is much improved. I have no further comment, and recommend acceptance.

We thank the reviewer for this positive appraisal.

Reviewer #3

The revisions and newly added analysis in the updated manuscript address my questions and comments. I only have two minor questions about the newly added parts, which in my opinion would help improve the clarity of the presentation of the revised manuscript:

We thank the reviewer for this positive appraisal of our work and constructive feedback. Our detailed responses are given below, interleaved with the reviewer's comments (in blue).

1) I think the fitting to the functional connectivity (Fig. 9) provides an important addition to the presented study and links it quite nicely to other studies on computational models in the literature. It is also quite important that the parameter combination yielding the smooth waves also have the second highest correlation across the entire parameter space. Can the authors please comment on the type of dynamics observed for the best fit between the model and the functional connectivity matrix, were they also wave dynamics?

The global best fit exhibited large-scale waves coexisting with discrete clusters. We now mention this on p20:

*only marginally lower than the best global fit (for $c=0.2$ and $\tau=0$ ms), **which exhibited large-scale waves coexisting with discrete clusters.***

Also if I understand correctly there is a typo in the newly added explanation about that: “Intriguingly we find that the smooth waves observed for the parameter combination of $c=0.6$ and $\tau=1$ ms yield the second highest correlation across the entire parameter space tested (with or without GSR), only marginally lower than the best global fit (for $c=0.6$ and $\tau=1$ ms).” I believe the values in the parenthesis should be the best global fit; i.e. $c=0.2$, $\tau=0$ ms.

Fixed.

2) The added analysis on the metastable transitions is also very intriguing. I think the additional finding showing that there is specific information in the inter-hemispheric connections is quite interesting. In my opinion, for the use of the partitioning in this study, the important point would be

that the detected metastable transitions (Fig. 3 b) are consistent across different partitions. Could the authors please comment on whether that is the case?

Any given partition opens a 'window' into the dynamics but doesn't necessarily capture all transitions. For example, a transition between two bilaterally symmetric patterns may not be visible using a left-right partition. We envisage a hierarchy of transitions: transitions for which the choice of spatial (or topological) partition matters, and more dramatic transitions that are visible in all partitions. Future work would be required to catalog these possibilities. We have added the following text on p8:

Thus some transitions occur across multiple partitions, while other partitions are sensitive to transitions with specific spatial (or topological) structure. This suggests a hierarchy of transitions: transitions for which the choice of spatial (or topological) partition matters, and more dramatic transitions that are evident in all partitions. Future work is required to catalog these possibilities.

Typo:

- Please define GSR in the manuscript.

Done (p20).